# Trading mental effort for confidence in the metacognitive control of value-based decision-making

**Douglas G Lee[1,2,3]\*, Jean Daunizeau[2,4]\***

[1]Sorbonne University, Paris, France; [2]Paris Brain Institute (ICM), Paris, France; [3]Institute of Cognitive Sciences and Technologies, National Research Council of Italy, Rome, Italy; [4]Translational Neuromodeling Unit (TNU), ETH, Zurich, Switzerland

**Abstract** Why do we sometimes opt for actions or items that we do not value the most? Under current neurocomputational theories, such preference reversals are typically interpreted in terms of errors that arise from the unreliable signaling of value to brain decision systems. But, an alternative explanation is that people may change their mind because they are reassessing the value of alternative options while pondering the decision. So, why do we carefully ponder some decisions, but not others? In this work, we derive a computational model of the metacognitive control of decisions or MCD. In brief, we assume that fast and automatic processes first provide initial (and largely uncertain) representations of options' values, yielding prior estimates of decision difficulty. These uncertain value representations are then refined by deploying cognitive (e.g., attentional, mnesic) resources, the allocation of which is controlled by an effort-confidence tradeoff. Importantly, the anticipated benefit of allocating resources varies in a decision-by-decision manner according to the prior estimate of decision difficulty. The ensuing MCD model predicts response time, subjective feeling of effort, choice confidence, changes of mind, as well as choice-induced preference change and certainty gain. We test these predictions in a systematic manner, using a dedicated behavioral paradigm. Our results provide a quantitative link between mental effort, choice confidence, and preference reversals, which could inform interpretations of related neuroimaging findings.

**\*For correspondence:**
DouglasGLee@gmail.com (DGL);
jean.daunizeau@gmail.com (JD)

**Competing interests:** The authors declare that no competing interests exist.

## Introduction

Why do we carefully ponder some decisions, but not others? Decisions permeate every aspect of our lives – what to eat, where to live, whom to date, etc. – but the amount of effort that we put into different decisions varies tremendously. Rather than processing all decision-relevant information, we often rely on fast habitual and/or intuitive decision policies, which can lead to irrational biases and errors (*Kahneman et al., 1982*). For example, snap judgments about others are prone to unconscious stereotyping, which often has enduring and detrimental consequences (*Greenwald and Banaji, 1995*). Yet we don't always follow the fast but negligent lead of habits or intuitions. So, what determines how much time and effort we invest when making decisions?

Biased and/or inaccurate decisions can be triggered by psychobiological determinants such as stress (*Porcelli and Delgado, 2009*; *Porcelli et al., 2012*), emotions (*Harlé and Sanfey, 2007*; *De Martino et al., 2006*; *Sokol-Hessner et al., 2013*), or fatigue (*Blain et al., 2016*). But, in fact, they also arise in the absence of such contextual factors. That is why they are sometimes viewed as the outcome of inherent neurocognitive limitations on the brain's decision processes, e.g., bounded attentional and/or mnemonic capacity (*Giguère and Love, 2013*; *Lim et al., 2011*; *Marois and Ivanoff, 2005*), unreliable neural representations of decision-relevant information (*Drugowitsch et al.,*

*2016*; *Wang and Busemeyer, 2016*; *Wyart and Koechlin, 2016*), or physiologically constrained neural information transmission (*Louie and Glimcher, 2012*; *Polanía et al., 2019*). However, an alternative perspective is that the brain has a preference for efficiency over accuracy (*Thorngate, 1980*). For example, when making perceptual or motor decisions, people frequently trade accuracy for speed, even when time constraints are not tight (*Heitz, 2014*; *Palmer et al., 2005*). Related neural and behavioral data are best explained by 'accumulation-to-bound' process models, in which a decision is emitted when the accumulated perceptual evidence reaches a bound (*Gold and Shadlen, 2007*; *O'Connell et al., 2012*; *Ratcliff and McKoon, 2008*; *Ratcliff et al., 2016*). Further computational work demonstrated that if the bound is properly set, these models actually implement an optimal solution to speed-accuracy tradeoff problems (*Ditterich, 2006*; *Drugowitsch et al., 2012*). From a theoretical standpoint, this implies that accumulation-to-bound policies can be viewed as an evolutionary adaptation, in response to selective pressure that favors efficiency (*Pirrone et al., 2014*).

This line of reasoning, however, is not trivial to generalize to value-based decision-making, for which objective accuracy remains an elusive notion (*Dutilh and Rieskamp, 2016*; *Rangel et al., 2008*). This is because, in contrast to evidence-based (e.g., perceptual) decisions, there are no right or wrong value-based decisions. Nevertheless, people still make choices that deviate from subjective reports of value, with a rate that decreases with value contrast. From the perspective of accumulation-to-bound models, these preference reversals count as errors and arise from the unreliable signaling of value to decision systems in the brain (*Lim et al., 2013*). That value-based variants of accumulation-to-bound models are able to capture the neural and behavioral effects of, e.g., overt attention (*Krajbich et al., 2010*; *Lim et al., 2011*), external time pressure (*Milosavljevic et al., 2010*), confidence (*De Martino et al., 2013*), or default preferences (*Lopez-Persem et al., 2016*) lends empirical support to this type of interpretation. Further credit also comes from theoretical studies showing that these process models, under some simplifying assumptions, optimally solve the problem of efficient value comparison (*Tajima et al., 2016*; *Tajima et al., 2019*). However, they do not solve the issue of adjusting the amount of effort to invest in reassessing an uncertain prior preference with yet-unprocessed value-relevant information. Here, we propose an alternative computational model of value-based decision-making that suggests that mental effort is optimally traded against choice confidence, given how value representations are modified while pondering decisions (*Slovic, 1995*; *Tversky and Thaler, 1990*; *Warren et al., 2011*).

We start from the premise that the brain generates representations of options' value in a quick and automatic manner, even before attention is engaged for comparing option values (*Lebreton et al., 2009*). The brain also encodes the certainty of such value estimates (*Lebreton et al., 2015*), from which a priori feelings of choice difficulty and confidence could, in principle, be derived. Importantly, people are reluctant to make a choice that they are not confident about (*De Martino et al., 2013*). Thus, when faced with a difficult decision, people should reassess option values until they reach a satisfactory level of confidence about their preference. This effortful mental deliberation would engage neurocognitive resources, such as attention and memory, in order to process value-relevant information. In line with recent proposals regarding the strategic deployment of cognitive control (*Musslick et al., 2015*; *Shenhav et al., 2013*), we assume that the amount of allocated resources optimizes a tradeoff between expected effort cost and confidence gain. The main issue here is that the impact of yet-unprocessed information on value representations is a priori unknown. Critically, we show how the system can anticipate the expected benefit of allocating resources before having processed value-relevant information. The ensuing *metacognitive control of decisions* or *MCD* thus adjusts mental effort on a decision-by-decision basis, according to prior decision difficulty and importance (*Figure 1*).

As we will see, the MCD model makes clear quantitative predictions about several key decision variables (cf. Model section below). We test these predictions by asking participants to report their judgments about each item's subjective value and their subjective certainty about their value judgments, both before and after choosing between pairs of the items. Note that we also measure choice confidence, response time, and subjective effort for each decision.

The objective of this work is to show how most non-trivial properties of value-based decision-making can be explained with a minimal (and mutually consistent) set of assumptions. The MCD model predicts response time, subjective effort, choice confidence, probability of changing one's mind, as well as choice-induced preference change and certainty gain, out of two properties of pre-

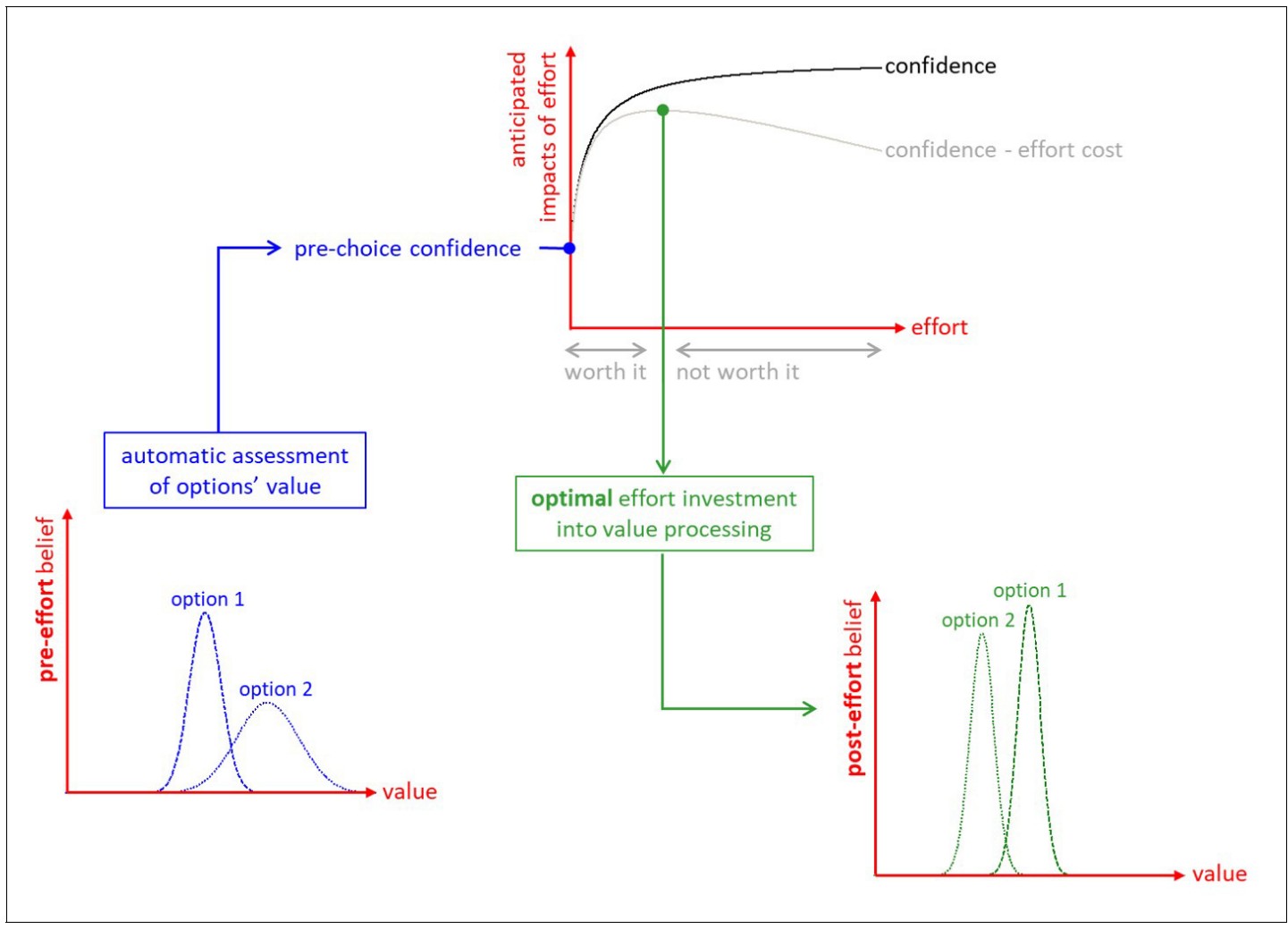

**Figure 1.** The metacognitive control of decisions. First, automatic processes provide a 'pre-effort' belief about option values. This belief is probabilistic, in the sense that it captures an uncertain prediction regarding the to-be-experienced value of a given option. This pre-effort belief serves to identify the anticipated impact of investing costly cognitive resources (i.e., effort) in the decision. In particular, investing effort is expected to increase decision confidence beyond its pre-effort level. But how much effort it should be worth investing depends upon the balance between expected confidence gain and effort costs. The system then allocates resources into value-relevant information processing up until the optimal effort investment is reached. At this point, a decision is triggered based on the current post-effort belief about option values (in this example, the system has changed its mind, i.e., its preference has reversed). Note: we refer to the ensuing increase in the value difference between chosen and unchosen items as the 'spreading of alternatives' (cf. Materials and methods section).

choice value representations, namely value ratings and value certainty ratings. Relevant details regarding the model derivations, as well as the decision-making paradigm we designed to evaluate those predictions, can be found in the Model and Methods sections below. In the subsequent section, we present our main dual computational/behavioral results. Finally, we discuss our results in light of the existing literature on value-based decision-making.

## The MCD model

In what follows, we derive a computational model of the metacognitive control of decisions or MCD. In brief, we assume that the amount of cognitive resources that is deployed during a decision is controlled by an effort-confidence tradeoff. Critically, this tradeoff relies on a prospective anticipation of how these resources will perturb the internal representations of subjective values. As we will see, the MCD model eventually predicts how cognitive effort expenditure depends upon prior estimates of decision difficulty, and what impact this will have on post-choice value representations.

## Deriving the expected value of decision control

Let $z$ be the amount of cognitive (e.g., executive, mnemonic, or attentional) resources that serve to process value-relevant information. Allocating these resources will be associated with both a benefit $B(z)$, and a cost $C(z)$. As we will see, both are increasing functions of $z$: $B(z)$ derives from the refinement of internal representations of subjective values of alternative options or actions that compose the choice set, and $C(z)$ quantifies how aversive engaging cognitive resources are (mental effort). In line with the framework of expected value of control or EVC (**Musslick et al., 2015**; **Shenhav et al., 2013**), we assume that the brain chooses to allocate the amount of resources $\hat{z}$ that optimizes the following cost–benefit trade-off:

$$\hat{z} = \arg\max_{z} E[B(z) - C(z)] \tag{1}$$

where the expectation accounts for predictable stochastic influences that ensue from allocating resources (this will be clarified below). Note that the benefit term $B(z)$ is the (weighted) choice confidence $P_c(z)$:

$$B(z) = R \times P_c(z) \tag{2}$$

where the weight $R$ is analogous to a reward and quantifies the importance of making a confident decision (see below). As we will see, $P_c(z)$ plays a pivotal role in the model, in that it captures the efficacy of allocating resources for processing value-relevant information. So, how do we define choice confidence?

We assume that decision makers may be unsure about how much they like/want the alternative options that compose the choice set. In other words, the internal representations of values $V_i$ of alternative options are probabilistic. Such a probabilistic representation of value can be understood in terms of, for example, an uncertain prediction regarding the to-be-experienced value of a given option. Without loss of generality, the probabilistic representation of option value takes the form of Gaussian probability density functions, as follows:

$$p(V_i) = N(\mu_i, \sigma_i) \tag{3}$$

where $\mu_i$ and $\sigma_i$ are the mode and the variance of the probabilistic value representation, respectively (and $i$ indexes alternative options in the choice set).

This allows us to define choice confidence $P_c$ as the probability that the (predicted) experienced value of the (to be) chosen item is higher than that of the (to be) unchosen item:

$$
\begin{aligned}
P_c &= \begin{cases} P(V_1 > V_2) \text{ if item \#1 is chosen} \\ P(V_2 > V_1) \text{ if item \#2 is chosen} \end{cases} \\
&= \begin{cases} P(V_1 > V_2) \text{ if } \Delta\mu > 0 \\ P(V_2 > V_1) \text{ if } \Delta\mu < 0 \end{cases} \\
&\approx s\left( \frac{\pi|\Delta\mu|}{\sqrt{3(\sigma_1 + \sigma_2)}} \right)
\end{aligned}
\tag{4}
$$

where $s(x) = 1/1 + e^{-x}$ is the standard sigmoid mapping. Here the second line derives from assuming that the choice follows the sign of the preference $\Delta\mu = \mu_1 - \mu_2$, and the last line derives from a moment-matching approximation to the Gaussian cumulative density function (**Daunizeau, 2017a**).

As stated in the Introduction section, we assume that the brain valuation system automatically generates uncertain estimates of options' value (**Lebreton et al., 2009**; **Lebreton et al., 2015**), before cognitive effort is invested in decision-making. In what follows, $\mu_i^0$ and $\sigma_i^0$ are the mode and variance of the ensuing prior value representations (we treat them as inputs to the MCD model). We also assume that these prior representations neglect existing value-relevant information that would require cognitive effort to be retrieved and processed (**Lopez-Persem et al., 2016**).

Now, how does the system anticipate the benefit of allocating resources to the decision process? Recall that the purpose of allocating resources is to process (yet unavailable) value-relevant information. The critical issue is thus to predict how both the uncertainty $\sigma_i$ and the modes $\mu_i$ of value representations will eventually change, before having actually allocated the resources (i.e., without having

processed the information). In brief, allocating resources essentially has two impacts: (i) it decreases the uncertainty $\sigma_i$, and (ii) it perturbs the modes $\mu_i$ in a stochastic manner.

The former impact derives from assuming that the amount of information that will be processed increases with the amount of allocated resources. Here, this implies that the variance of a given probabilistic value representation decreases in proportion to the amount of allocated effort, that is:

$$\sigma_i \triangleq \sigma_i(z) = \frac{1}{\frac{1}{\sigma_i^0} + \beta z} \tag{5}$$

where $\sigma_i^0$ is the prior variance of the representation (before any effort has been allocated), and $\beta$ controls the efficacy with which resources increase the precision of the value representation. Formally speaking, *Equation 5* has the form of a Bayesian update of the belief's precision in a Gaussian-likelihood model, where the precision of the likelihood term is $\beta z$. More precisely, $\beta$ is the precision increase that follows from allocating a unitary amount of resources $z$. In what follows, we will refer to $\beta$ as the '*type #1 effort efficacy*'.

The latter impact follows from acknowledging the fact that the system cannot know how processing more value-relevant information will affect its preference before having allocated the corresponding resources. Let $\delta_i(z)$ be the change in the position of the mode of the $i$th value representation, having allocated an amount $z$ of resources. The direction of the mode's perturbation $\delta_i(z)$ cannot be predicted because it is tied to the information that would be processed. However, a tenable assumption is to consider that the magnitude of the perturbation increases with the amount of information that will be processed. This reduces to stating that the variance of $\delta_i(z)$ increases in proportion to $z$, that is:

$$\mu_i(z) = \mu_i^0 + \delta_i$$
$$\delta_i \sim N(0, \gamma z) \tag{6}$$

where $\mu_i^0$ is the mode of the value representation before any effort has been allocated, and $\gamma$ controls the relationship between the amount of allocated resources and the variance of the perturbation term $\delta$. The higher the parameter $\gamma$, the greater the expected perturbation of the mode for a given amount of allocated resources. In what follows, we will refer to $\gamma$ as the '*type #2 effort efficacy*'. Note that *Equation 6* treats the impact of future information processing as a non-specific random perturbation on the mode of the prior value representation. Our justification for this assumption is twofold: (i) it is simple, and (ii) and it captures the idea that the MCD controller does not know how the allocated resources will be used (here, by the value-based decision system downstream). We will see that, in spite of this, the MCD controller can still make quantitative predictions regarding the expected benefit of allocating resources.

Taken together, Equations 5 and 6 imply that predicting the net effect of allocating resources onto choice confidence is not trivial. On one hand, allocating effort will increase the precision of value representations (cf. *Equation 5*), which mechanically increases choice confidence, all other things being equal. On the other hand, allocating effort can either increase or decrease the absolute difference $|\Delta\mu(z)|$ between the modes. This, in fact, depends upon the sign of the perturbation terms $\delta$, which are not known in advance. Having said this, it is possible to derive the *expected* absolute difference between the modes that would follow from allocating an amount $z$ of resources:

$$E[|\Delta\mu| \mid z] = 2\sqrt{\frac{\gamma z}{\pi}} exp\left(-\frac{|\Delta\mu^0|^2}{4\gamma z}\right) + \Delta\mu^0\left(2 \times s\left(\frac{\pi\Delta\mu^0}{\sqrt{6\gamma z}}\right) - 1\right) \tag{7}$$

where we have used the expression for the first-order moment of the so-called 'folded normal distribution', and the second term in the right-hand side of *Equation 7* derives from the same moment-matching approximation to the Gaussian cumulative density function as above. The expected absolute means' difference $E[|\Delta\mu| \mid z]$ depends upon both the absolute prior mean difference $|\Delta\mu^0|$ and the amount of allocated resources $z$. This is depicted in *Figure 2*.

One can see that $E[|\Delta\mu| \mid z] - |\Delta\mu^0|$ is always greater than 0 and increases with $z$ (and if $z = 0$, then $E[|\Delta\mu| \mid z] = \Delta\mu^0|$). In other words, allocating resources is expected to increase the value difference, despite the fact that the impact of the perturbation term can go either way. In addition, the

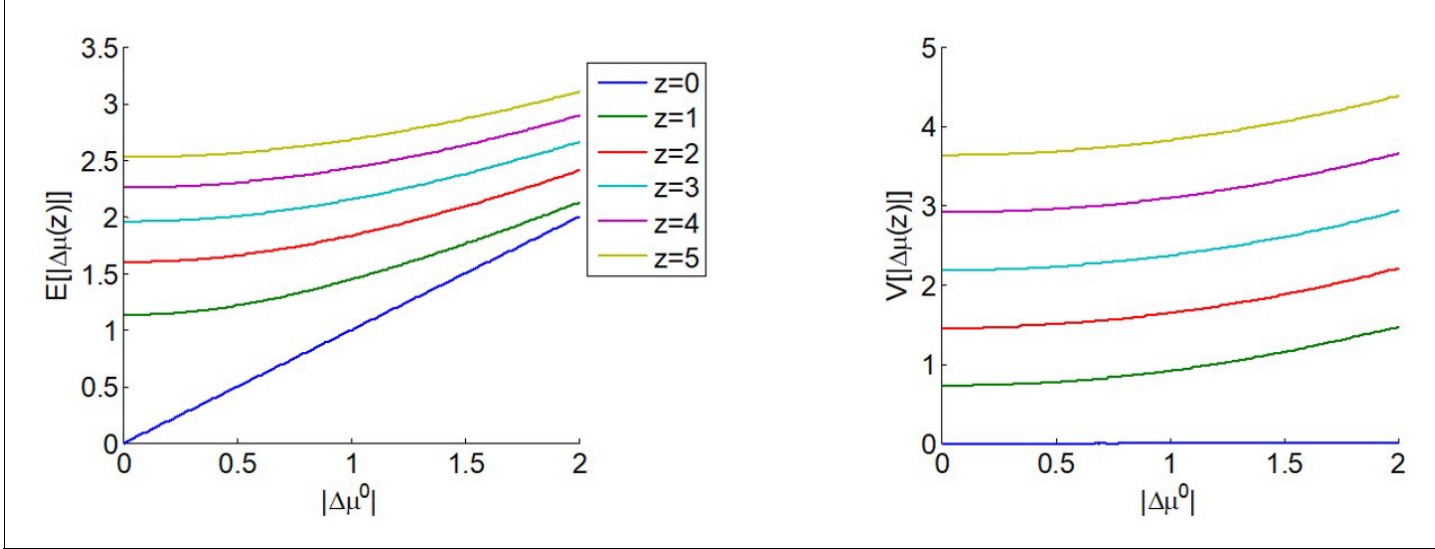

**Figure 2.** The expected impact of allocated resources onto value representations. Left panel: the expected absolute mean difference $E\left[|\Delta\mu(z)|\,|z\right]$ (y-axis) is plotted as a function of the absolute prior mean difference $|\mu^0|$ (x-axis) for different amounts $z$ of allocated resources (color code), having set type #2 effort efficacy to unity (i.e. $\gamma = 1$). Right panel: Variance $V\left[|\Delta\mu(z)|\,|z\right]$ of the absolute mean difference; same format.

expected gain in value difference afforded by allocating resources decreases with the absolute prior means' difference.

Similarly, the variance $V[|\Delta\mu|\,|z]$ of the absolute means' difference is derived from the expression of the second-order moment of the corresponding folded normal distribution:

$$V[|\Delta\mu|\,|z] = 2\gamma z + |\Delta\mu^0|^2 - E[|\Delta\mu|\,|z]^2 \tag{8}$$

One can see in *Figure 2* that $V[|\Delta\mu|\,|z]$ increases with the amount $z$ of allocated resources (but if $z = 0$, then $V[|\Delta\mu|\,|z] = 0$).

Knowing the moments of the distribution of $|\Delta\mu|$ now enables us to derive the expected confidence level $\bar{P}_c(z)$ that would result from allocating the amount of resource $z$:

$$
\begin{aligned}
\bar{P}_c(z) &\triangleq E[P_c|z] \\
&= E\left[s\left(\frac{\pi|\Delta\mu|}{\sqrt{6\sigma(z)}}\right)|z\right] \\
&\approx s\left(\frac{\pi E[|\Delta\mu|\,|z]}{\sqrt{6\left(\sigma(z) + \frac{1}{2}V[|\Delta\mu|\,|z]\right)}}\right)
\end{aligned} \tag{9}
$$

where we have assumed, for the sake of conciseness, that both prior value representations are similarly uncertain (i.e., $\sigma_1^0 \approx \sigma_2^0 \triangleq \sigma^0$). It turns out that the expected choice confidence $\bar{P}_c(z)$ always increase with $z$, irrespective of the efficacy parameters, as long as $\beta \neq 0$ or $\gamma \neq 0$. These, however, control the magnitude of the confidence gain that can be expected from allocating an amount $z$ of resources. *Equation 9* is important, because it quantifies the expected benefit of resource allocation, before having processed the ensuing value-relevant information. More details regarding the accuracy of *Equation 9* can be found in section 1 of the Appendix. In addition, section 2 of the Appendix summarizes the dependence of MCD-optimal choice confidence on $|\Delta\mu^0|$ and $\sigma^0$.

To complete the cost–benefit model, we simply assume that the cost of allocating resources to the decision process linearly scales with the amount of resources, that is:

$$C(z) = \alpha z \tag{10}$$

where $\alpha$ determines the effort cost of allocating a unitary amount of resources $z$. In what follows, we

will refer to $\alpha$ as the 'effort unitary cost'. We note that weak nonlinearities in the cost function (e.g., quadratic terms) would not qualitatively change the model predictions.

In brief, the MCD-optimal resource allocation $\hat{z} \triangleq \hat{z}(\alpha, \beta, \gamma)$ is simply given by:

$$\hat{z} = \arg\max_z [R \times \bar{P}_c(z) - \alpha z] \tag{11}$$

which does not have any closed-form analytic solution. Nevertheless, it can easily be identified numerically, having replaced *Equations 7–9* into *Equation 11*. We refer the readers interested in the impact of model parameters $\{\alpha, \beta, \gamma\}$ on the MCD-optimal control to section 2 of the Appendix.

At this point, we do not specify how *Equation 11* is solved by neural networks in the brain. Many alternatives are possible, from gradient ascent (*Seung, 2003*) to winner-take-all competition of candidate solutions (*Mao and Massaquoi, 2007*). We will also comment on the specific issue of prospective (offline) versus reactive (online) MCD processes in the Discussion section.

*Note*: implicit in the above model derivation is the assumption that the allocation of resources is similar for both alternative options in the choice set (i.e. $z_1 \approx z_2 \triangleq z$). This simplifying assumption is justified by eye-tracking data (cf. section 8 of the Appendix).

## Corollary predictions of the MCD model

In the previous section, we derived the MCD-optimal resource allocation $\hat{z}$, which effectively best balances the expected choice confidence with the expected effort costs, given the predictable impact of stochastic perturbations that arise from processing value-relevant information. This quantitative prediction is effectively shown in Figures 5 and 6 of the Results section below, as a function of (empirical proxies for) the prior absolute difference between modes $|\Delta\mu^0|$ and the prior certainty $1/\sigma^0$ of value representations. But, this resource allocation mechanism has a few interesting corollary implications.

To begin with, note that knowing $\hat{z}$ enables us to predict what confidence level the system should eventually reach. In fact, one can define the MCD-optimal confidence level as the expected confidence evaluated at the MCD-optimal amount of allocated resources, that is, $\bar{P}_c(\hat{z})$. This is important, because it implies that the model can predict both the effort the system will invest and its associated confidence, on a decision-by-decision basis. The impact of the efficacy parameters on this quantitative prediction is detailed in section 2 of the Appendix.

Additionally, $\hat{z}$ determines the expected improvement in the certainty of value representations (hereafter: the 'certainty gain'), which trivially relates to type #2 efficacy, that is: $1/\sigma(\hat{z}) - 1/\sigma^0 = \beta\hat{z}$. This also means that, under the MCD model, no choice-induced value certainty gain can be expected when $\beta = 0$.

Similarly, one can predict the MCD-optimal probability of changing one's mind. Recall that the probability $Q(z)$ of changing one's mind depends on the amount of allocated resources $z$, that is:

$$\begin{aligned}
Q(z) &\triangleq P\big(sign(\Delta\mu) \neq sign(\Delta\mu^0)|z\big) \\
&= \begin{cases} cP(\Delta\mu>0|z) \text{ if } \Delta\mu^0<0 \\ P(\Delta\mu<0|z) \text{ if } \Delta\mu^0>0 \end{cases} \\
&\approx s\left(-\frac{\pi|\Delta\mu^0|}{\sqrt{6\gamma z}}\right)
\end{aligned} \tag{12}$$

One can see that the MCD-optimal probability of changing one's mind $Q(\hat{z})$ is a simple monotonic function of the allocated effort $\hat{z}$. Importantly, $Q(z) = 0$ when $\gamma = 0$. This implies that MCD agents do not change their minds when effort cannot change the relative position of the modes of the options' value representations (irrespective of type #1 effort efficacy). In retrospect, this is critical, because there should be no incentive to invest resources in deliberation if it would not be possible to change one's pre-deliberation preference.

Lastly, we can predict the magnitude of choice-induced preference change, that is, how value representations are supposed to spread apart during the decision. Such an effect is typically measured in terms of the so-called 'spreading of alternatives' or SoA, which is defined as follows:

$$SOA = \left(\mu_{chosen}^{(post-choice)} - \mu_{unchosen}^{(post-choice)}\right) - \left(\mu_{chosen}^{(pre-choice)} - \mu_{unchosen}^{(pre-choice)}\right)$$
$$= \begin{cases} c\Delta\mu(z) - \Delta\mu^0 & \text{if } \Delta\mu(z) > 0 \\ \Delta\mu^0 - \Delta\mu(z) & \text{if } \Delta\mu(z) < 0 \end{cases} \quad (13)$$
$$= \begin{cases} c\Delta\delta(z) & \text{if } \Delta\delta(z) > -\Delta\mu^0 \\ -\Delta\delta(z) & \text{if } \Delta\delta(z) < -\Delta\mu^0 \end{cases}$$

where $\Delta\delta(z) \sim N(0, 2\gamma z)$ is the cumulative perturbation term of the modes' difference. Taking the expectation of the right-hand term of **Equation 13** under the distribution of $\Delta\delta(z)$ and evaluating it at $z = \hat{z}$ now yields the MCD-optimal spreading of alternatives $\bar{SOA}(\hat{z})$:

$$\bar{SOA}(\hat{z}) = E[SOA|\hat{z}]$$
$$= E\left[\Delta\delta(\hat{z})|\Delta\delta(\hat{z}) > -\Delta\mu^0\right]P\left(\Delta\delta(\hat{z}) > -\Delta\mu^0\right)$$
$$- E\left[\Delta\delta(\hat{z})|\Delta\delta(\hat{z}) < -\Delta\mu^0\right]P\left(\Delta\delta(\hat{z}) < -\Delta\mu^0\right) \quad (14)$$
$$= 2\sqrt{\frac{\gamma\hat{z}}{\pi}}\exp\left(-\frac{|\Delta\mu^0|^2}{4\gamma\hat{z}}\right)$$

where the last line derives from the expression of the first-order moment of the truncated Gaussian distribution. Note that the expected preference change also increases monotonically with the allocated effort $\hat{z}$. Here again, under the MCD model, no preference change can be expected when $\gamma = 0$.

We note that all of these corollary predictions essentially capture choice-induced modifications of value representations. This is why we will refer to choice confidence, value certainty gain, change of mind, and spreading of alternatives as 'decision-related' variables.

## Correspondence between model variables and empirical measures

In summary, the MCD model predicts cognitive effort (or, more properly, the amount of allocated resources) and decision-related variables, given the prior absolute difference between modes $|\Delta\mu^0|$ and the prior certainty $1/\sigma^0$ of value representations. In other words, the inputs to the MCD model are the prior moments of value representations, whose trial-by-trial variations determine variations in model predictions. Here, we simply assume that pre-choice value and value certainty ratings provide us with an approximation of these prior moments. More precisely, we use $\Delta VR^0$ and $VCR^0$ (cf. section 3.3 below) as empirical proxies for $\Delta\mu^0$ and $1/\sigma^0$, respectively. Accordingly, we consider post-choice value and value certainty ratings as empirical proxies for the posterior mean $\mu(\hat{z})$ and precision $1/\sigma(\hat{z})$ of value representations, at the time when the decision was triggered (i.e., after having invested the effort $\hat{z}$). Similarly, we match expected choice confidence $\bar{P}_c(z)$ (i.e., after having invested the effort $\hat{z}$) with empirical choice confidence.

Note that the MCD model does not specify *what* the allocated resources are. In principle, both mnesic and attentional resources may be engaged when processing value-relevant information. Nevertheless, what really matters is assessing the magnitude $z$ of decision-related effort. We think of $z$ as the cumulative engagement of neurocognitive resources, which varies both in terms of duration and intensity. Empirically, we relate $\hat{z}$ to two different 'effort-related' empirical measures, namely subjective feeling of effort and response time. The former relies on the subjective cost incurred when deploying neurocognitive resources, which would be signaled by experiencing mental effort. The latter makes sense if one thinks of response time in terms of effort duration. Although it is a more objective measurement than subjective rating of effort, response time only approximates $\hat{z}$ if effort intensity shows relatively small variations. We will comment on this in the Discussion section.

Finally, the MCD model is also agnostic about the definition of 'decision importance', that is, the weight $R$ in **Equation 2**. In this work, we simply investigate the effect of decision importance by comparing subjective effort and response time in 'neutral' versus 'consequential' decisions (cf. section 'Task conditions' below). We will also comment on this in the Discussion section.

# Materials and methods

## Participants

Participants for our study were recruited from the RISC (*Relais d'Information sur les Sciences de la Cognition*) subject pool through the ICM (*Institut du Cerveau et de la Moelle – Paris Brain Institute*). All participants were native French speakers, with no reported history of psychiatric or neurological illness. A total of 41 people (28 female; age: mean = 28, SD = 5, min = 20, max = 40) participated in this study. The experiment lasted approximately 2 hr, and participants were paid a flat rate of 20€ as compensation for their time, plus a bonus, which was given to participants to compensate for potential financial losses in the 'penalized' trials (see below). More precisely, in 'penalized' trials, participants lost 0.20€ (out of a 5€ bonus) for each second that they took to make their choice. This resulted in an average 4€ bonus (across participants). One group of 11 participants was excluded from the cross-condition analysis only (see below) due to technical issues.

## Materials

Written instructions provided detailed information about the sequence of tasks within the experiment, the mechanics of how participants would perform the tasks, and images illustrating what a typical screen within each task section would look like. The experiment was developed using Matlab and PsychToolbox, and was conducted entirely in French. The stimuli for this experiment were 148 digital images, each representing a distinct food item (50 fruits, 50 vegetables, and 48 various snack items including nuts, meats, and cheeses). Food items were selected such that most items would be well known to most participants.

Eye gaze position and pupil size were continuously recorded throughout the duration of the experiment using The Eye Tribe eye-tracking devices. Participants' head positions were fixed using stationary chinrests. In case of incidental movements, we corrected the pupil size data for distance to screen, separately for each eye.

## Task design

Prior to commencing the testing session of the experiment, participants underwent a brief training session. The training tasks were identical to the experimental tasks, although different stimuli were used (beverages). The experiment itself began with an initial section where all individual items were displayed in a random sequence for 1.5 s each, in order to familiarize the participants with the set of options they would later be considering and form an impression of the range of subjective value for the set. The main experiment was divided into three sections, following the classic Free-Choice Paradigm protocol (e.g., *Izuma and Murayama, 2013*): pre-choice item ratings, choice, and post-choice item ratings. There was no time limit for the overall experiment, nor for the different sections, nor for the individual trials. The item rating and choice sections are described below (see *Figure 3*).

### Item rating (same for pre-choice and post-choice sessions)

Participants were asked to rate the entire set of items in terms of how much they liked each item. The items were presented one at a time in a random sequence (pseudo-randomized across participants). At the onset of each trial, a fixation cross appeared at the center of the screen for 750 ms. Next, a solitary image of a food item appeared at the center of the screen. Participants had to respond to the question, 'How much do you like this item?' using a horizontal slider scale (from 'I hate it!' to 'I love it!') to indicate their value rating for the item. The middle of the scale was the point of neutrality ('I don't care about it."). Hereafter, we refer to the reported value as the 'pre-choice value rating'. Participants then had to respond to the question, 'What degree of certainty do you have?' (about the item's value) by expanding a solid bar symmetrically around the cursor of the value slider scale to indicate the range of possible value ratings that would be compatible with their subjective feeling. We measured participants' certainty about value rating in terms of the percentage of the value scale that is not occupied by the reported range of compatible value ratings. We refer to this as the 'pre-choice value certainty rating'. At that time, the next trial began.

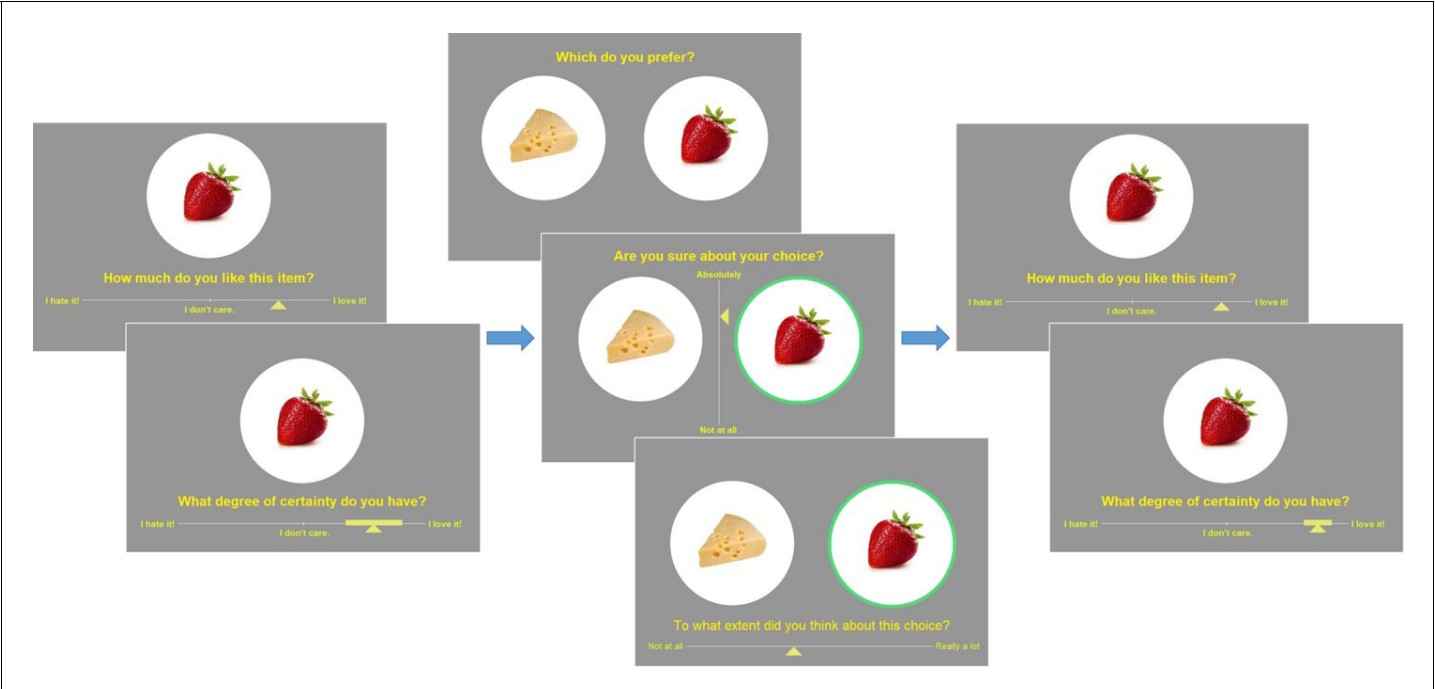

**Figure 3.** Experimental design. Left: pre-choice item rating session: participants are asked to rate how much they like each food item and how certain they are about it (value certainty rating). Center: choice session: participants are asked to choose between two food items, to rate how confident they are about their choice, and to report the feeling of effort associated with the decision. Right: post-choice item rating session (same as pre-choice item rating session).

## Note

In the Results section below, $\Delta VR^0$ is the difference between pre-choice value ratings of items composing a choice set. Similarly, $VCR^0$ is the average pre-choice value certainty ratings across items composing a choice set. Both value and value certainty rating scales range from 0 to 1 (but participants were unaware of the quantitative units of the scales).

## Choice

Participants were asked to choose between pairs of items in terms of which item they preferred. The entire set of items was presented one pair at a time in a random sequence. Each item appeared in only one pair. At the onset of each trial, a fixation cross appeared at the center of the screen for 750 ms. Next, two images of snack items appeared on the screen: one toward the left and one toward the right. Participants had to respond to the question, 'Which do you prefer?' using the left or right arrow key. We measured response time in terms of the delay between the stimulus onset and the response. Participants then had to respond to the question, 'Are you sure about your choice?' using a vertical slider scale (from 'Not at all!' to 'Absolutely!'). We refer to this as the report of choice confidence. Finally, participants had to respond to the question, 'To what extent did you think about this choice?' using a horizontal slider scale (from 'Not at all!' to 'Really a lot!"). We refer to this as the report of subjective effort. At that time, the next trial began.

### Task conditions

We partitioned the task trials into three conditions, which were designed to test the following two predictions of the MCD model: all else equal, effort should increase with decision importance and decrease with related costs. We aimed to check the former prediction by asking participants to make some decisions where they knew that the choice would be real, that is, they would actually have to eat the chosen food item at the end of the experiment. We refer to these trials as 'consequential' decisions. To check the latter prediction, we imposed a financial penalty that increased with response time. More precisely, participants were instructed that they would lose 0.20€ (out of a

5€ bonus) for each second that they would take to make their choice. The choice section of the experiment was composed of 60 'neutral' trials, 7 'consequential' trials, and 7 'penalized' trials, which were randomly intermixed. Instructions for both 'consequential' and 'penalized' decisions were repeated at each relevant trial, immediately prior to the presentation of the choice items.

## Probabilistic model fit

The MCD model predicts trial-by-trial variations in the probability of changing one's mind, choice confidence, spreading of alternatives, certainty gain, response time, and subjective effort ratings (MCD outputs) from trial-by-trial variations in value rating difference $\Delta VR^0$ and mean value certainty rating $VCR^0$ (MCD inputs). Together, three unknown parameters control the quantitative relationship between MCD inputs and outputs: the *effort unitary cost* $\alpha$, *type #1 effort efficacy* $\beta$, and *type #2 effort efficacy* $\gamma$. However, additional parameters are required to capture variations induced by experimental conditions. Recall that we expect 'consequential' decisions to be more important than 'neutral' ones, and 'penalized' decisions effectively include an extraneous cost-of-time term. One can model the former condition effect by making $R$ (cf. *Equation 2*) sensitive to whether the decision is consequential or not. We proxy the latter condition effect by making the effort unitary cost $\alpha$ a function of whether the decision is penalized (where effort induces both intrinsic and extrinsic costs) or not (intrinsic effort cost only). This means that condition effects require one additional parameter each.

In principle, all of these parameters may vary across people, thereby capturing idiosyncrasies in people's (meta-)cognitive apparatus. However, in addition to estimating these five parameters, fitting the MCD model to each participant's data also requires a rescaling of the model's output variables. This is because there is no reason to expect the empirical measure of these variables to match their theoretical scale. We thus inserted two additional nuisance parameters per output MCD variable, which operate a linear rescaling (affine transformation, with a positive constraint on slope parameters). Importantly, these nuisance parameters cannot change the relationship between MCD inputs and outputs. In other terms, the MCD model really has only five degrees of freedom.

For each subject, we fit all MCD dependent variables concurrently with a single set of MCD parameters. Within-subject probabilistic parameter estimation was performed using the variational Laplace approach (*Daunizeau, 2017b*; *Friston et al., 2007*), which is made available from the VBA toolbox (*Daunizeau et al., 2014*). We refer the reader interested in the mathematical details of within-subject MCD parameter estimation to the section 3 of the Appendix (this also includes a parameter recovery analysis). In what follows, we compare empirical data to MCD-fitted dependent variables (when binned according to $\Delta VR^0$ and $VCR^0$). We refer to the latter as 'postdictions', in the sense that they derive from a posterior predictive density that is conditional on the corresponding data.

We also fit the MCD model on reduced subsets of dependent variables (e.g., only 'effort-related' variables), and report proper out-of-sample predictions of data that were not used for parameter estimation (e.g., 'decision-related' variables). We note that this is a strong test of the model, since it does not rely on any train/test partition of the predicted variable (see next section below).

## Results

Here, we test the predictions of the MCD model. We note that basic descriptive statistics of our data, including measures of test–retest reliability and replications of previously reported effects on confidence in value-based choices (*De Martino et al., 2013*), are appended in sections 5–7 of the Appendix.

### Within-subject model fit accuracy and out-of-sample predictions

To capture idiosyncrasies in participants' metacognitive control of decisions, the MCD model was fitted to subject-specific trial-by-trial data, where all MCD outputs (namely change of mind, choice confidence, spreading of alternatives, value certainty gain, response time, and subjective effort ratings) were considered together. In the next section, we present summary statistics at the group level, which validate the predictions that can be derived from the MCD model, when fitted to all dependent variables. But can we provide even stronger evidence that the MCD model is capable of predicting all dependent variables at once? In particular, can the model make out-of-sample

predictions regarding effort-related variables (i.e., RT and subjective effort ratings) given decision-related variables (i.e., choice confidence, change of mind, spreading of alternatives, and certainty gain), and *vice versa*?

To address this question, we performed two partial model fits: (i) with decision-related variables only, and (ii) with effort-related variables only. In both cases, out-of-sample predictions for the remaining dependent variables were obtained directly from within-subject parameter estimates. For each subject, we then estimated the cross-trial correlation between each pair of observed and predicted variables. *Figure 4* reports the ensuing group-average correlations, for each dependent variable and each model fit. In this context, the predictions derived when fitting the full dataset only serve as a reference point for evaluating the accuracy of out-of-sample predictions. For completeness, we also show chance-level prediction accuracy (i.e. the 95% percentile of group average correlations between observed and predicted variables under the null).

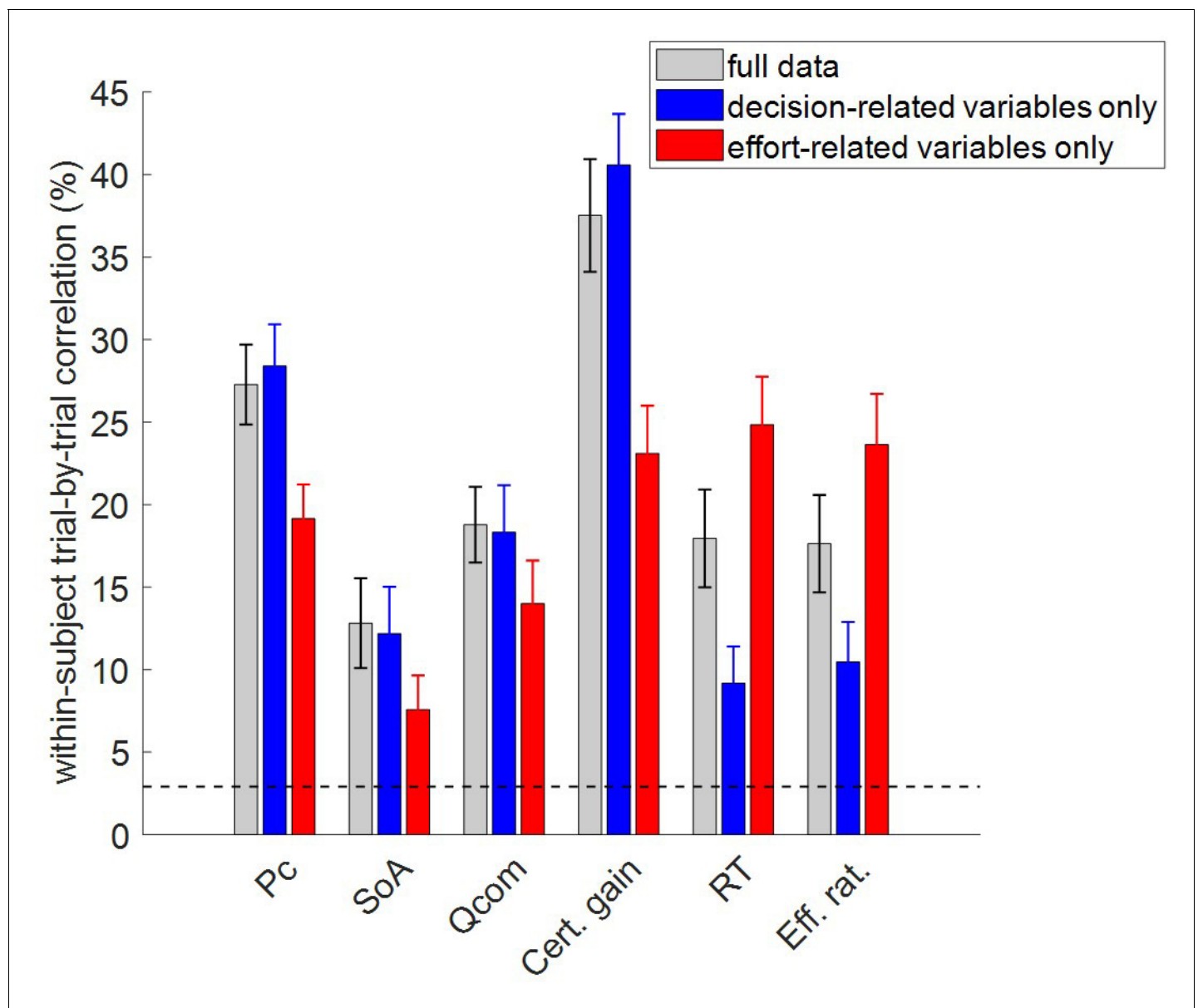

**Figure 4.** Accuracy of model postdictions and out-of-sample predictions. The mean within-subject (across-trial) correlation between observed and predicted/postdicted data (y-axis) is plotted for each variable (x-axis, from left to right: choice confidence, spreading of alternatives, change of mind, certainty gain, RT and subjective effort ratings), and each fitting procedure (gray: full data fit, blue: decision-related variables only, and red: effort-related variables only). Error bars depict standard error of the mean, and the horizontal dashed black line shows chance-level prediction accuracy.

In what follows, we refer to model predictions on dependent variables that were actually fitted by the model as 'postdictions' (full data fits: all dependent variables, partial model fits: variables included in the fit). As one would expect, the accuracy of postdictions is typically higher than that of out-of-sample predictions. Slightly more interesting, perhaps, is the fact that the accuracy of model predictions/postdictions depends upon which output variable is considered. For example, choice confidence is always better predicted/postdicted than spreading of alternatives. This is most likely because the latter data has lower reliability.

But the main result of this analysis is the fact that out-of-sample predictions of dependent variables perform systematically better than chance. In fact, all across-trial correlations between observed and predicted (out-of-sample) data were statistically significant at the group-level (all $p < 10^{-3}$). In particular, this implies that the MCD model makes accurate out-of-sample predictions regarding effort-related variables given decision-related variables, and reciprocally.

## Predicting effort-related variables

In what follows, we inspect the three-way relationships between pre-choice value and value certainty ratings and each effort-related variable: namely, RT and subjective effort rating. The former can be thought of as a proxy for the duration of resource allocation, whereas the latter is a metacognitive readout of resource allocation cost. Unless stated otherwise, we will focus on both the absolute difference between pre-choice value ratings (hereafter: $|\Delta VR^0|$) and the mean pre-choice value certainty rating across paired choice items (hereafter: $VCR^0$). Under the MCD model, increasing $|\Delta VR^0|$ and/or $VCR^0$ will decrease the demand for effort, which should result in smaller expected RT and subjective effort rating. We will now summarize the empirical data and highlight the corresponding quantitative MCD model postdictions and out-of-sample predictions (here: predictions are derived from model fits on decision-related variables only, that is, all dependent variables except RT and subjective effort rating).

First, we checked how RT relates to pre-choice value and value certainty ratings. For each subject, we regressed (log-) RT data against $|\Delta VR^0|$ and $VCR^0$, and then performed a group-level random-effect analysis on regression weights. The results of this model-free analysis provide a qualitative summary of the impact of trial-by-trial variations in pre-choice value representations on RT. We also compare RT data with both MCD model postdictions (full data fit) and out-of-sample predictions. In addition to summarizing the results of the model-free analysis, *Figure 5* shows the empirical, predicted, and postdicted RT data, when median-split (within subjects) according to both $|\Delta VR^0|$ and $VCR^0$.

One can see that RT data behave as expected under the MCD model, that is, RT decreases when $|\Delta VR^0|$ and/or $VCR^0$ increases. The random effect analysis shows that both variables have a significant negative effect at the group level ($|\Delta VR^0|$: mean standardized regression weight = $-0.16$, s.e.m. = 0.02, $p < 10^{-3}$; $VCR^0$: mean standardized regression weight = $-0.08$, s.e.m. = 0.02, $p < 10^{-3}$; one-sided t-tests). Moreover, MCD postdictions are remarkably accurate at capturing the effect of both $|\Delta VR^0|$ and $VCR^0$ variables in a quantitative manner. Although MCD out-of-sample predictions are also very accurate, they tend to slightly underestimate the quantitative effect of $|\Delta VR^0|$. This is because this effect is typically less pronounced in decision-related variables than in effort-related variables (see below), which then yield MCD parameter estimates that eventually attenuate the impact of $|\Delta VR^0|$ on effort.

Second, we checked how subjective effort ratings relate to pre-choice value and value certainty ratings. We performed the same analysis as above, the results of which are summarized in *Figure 6*.

Here as well, subjective effort rating data behave as expected under the MCD model, that is, subjective effort decreases when $|\Delta VR^0|$ and/or $VCR^0$ increases. The random effect analysis shows that both variables have a significant negative effect at the group level ($|\Delta VR^0|$: mean standardized regression weight = $-0.21$, s.e.m. = 0.03, $p < 10^{-3}$; $VCR^0$: mean regression weight = $-0.05$, s.e.m. = 0.02, $p = 0.027$; one-sided t-tests). One can see that MCD postdictions and out-of-sample predictions accurately capture the effect of both $|\Delta VR^0|$ and $VCR^0$ variables. More quantitatively, we note that MCD postdictions slightly overestimate the effect $VCR^0$, whereas out-of-sample predictions also tend to underestimate the effect of $|\Delta VR^0|$.

At this point, we note that the MCD model makes two additional predictions regarding effort-related variables, which relate to our task conditions. In brief, all else equal, effort should increase in 'consequential' trials, while it should decrease in 'penalized' trials. To test these predictions, we

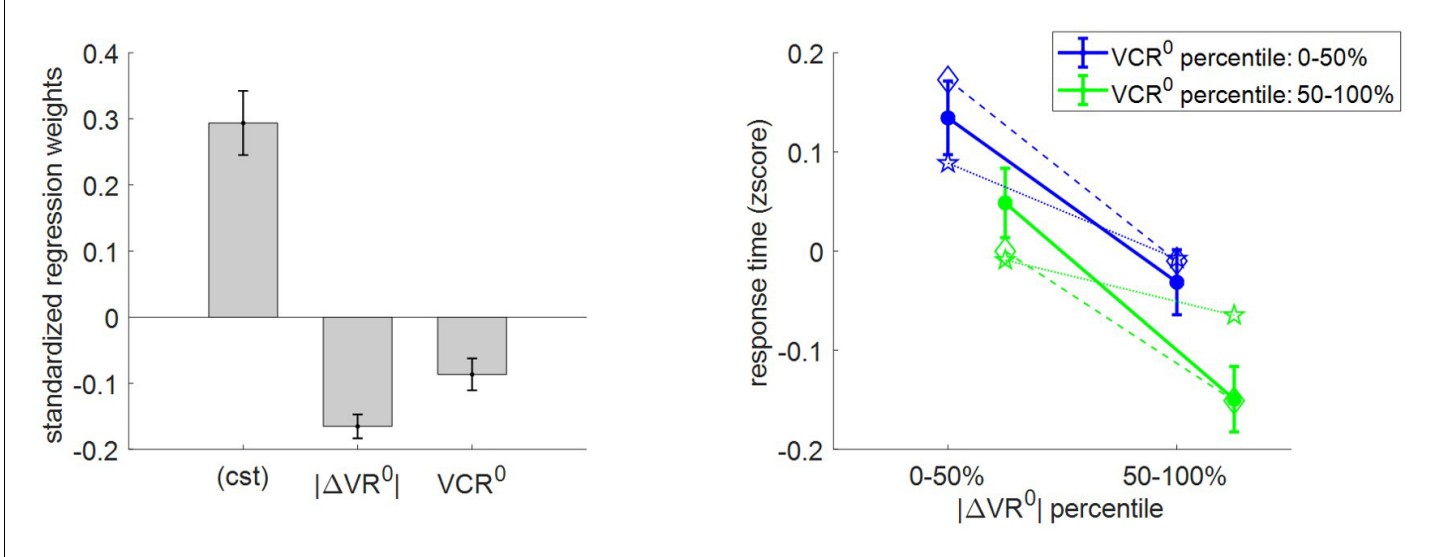

**Figure 5.** Three-way relationship between RT, value, and value certainty. Left panel: Mean standardized regression weights for $|\Delta VR^0|$ and $VCR^0$ on log-RT (*cst* is the constant term); error bars represent s.e.m. Right panel: Mean z-scored log-RT (y-axis) is shown as a function of $|\Delta VR^0|$ (x-axis) and $VCR^0$ (color code: blue = 0–50% lower quantile, green = 50–100% upper quantile); solid lines indicate empirical data (error bars represent s.e.m.), star-dotted lines show out-of-sample predictions and diamond-dashed lines represent model postdictions.

modified the model-free regression analysis of RT and subjective effort ratings by including two additional subject-level regressors, encoding consequential and penalized trials, respectively. *Figure 7* shows the ensuing augmented set of standardized regression weights for both RT and subjective effort ratings.

First, we note that accounting for task conditions does not modify the statistical significance of the impact of $|\Delta VR^0|$ and $VCR^0$ on effort-related variables, except for the effect of $VCR^0$ on subjective effort ratings (p=0.09, one-sided t-test). Second, one can see that the impact of 'consequential' and 'penalized' conditions on effort-related variables globally conforms to MCD predictions. More precisely, both RT and subjective effort ratings were significantly higher for 'consequential' decisions than for 'neutral' decisions (log-RT: mean standardized regression weight = 0.07, s.e.m. = 0.03,

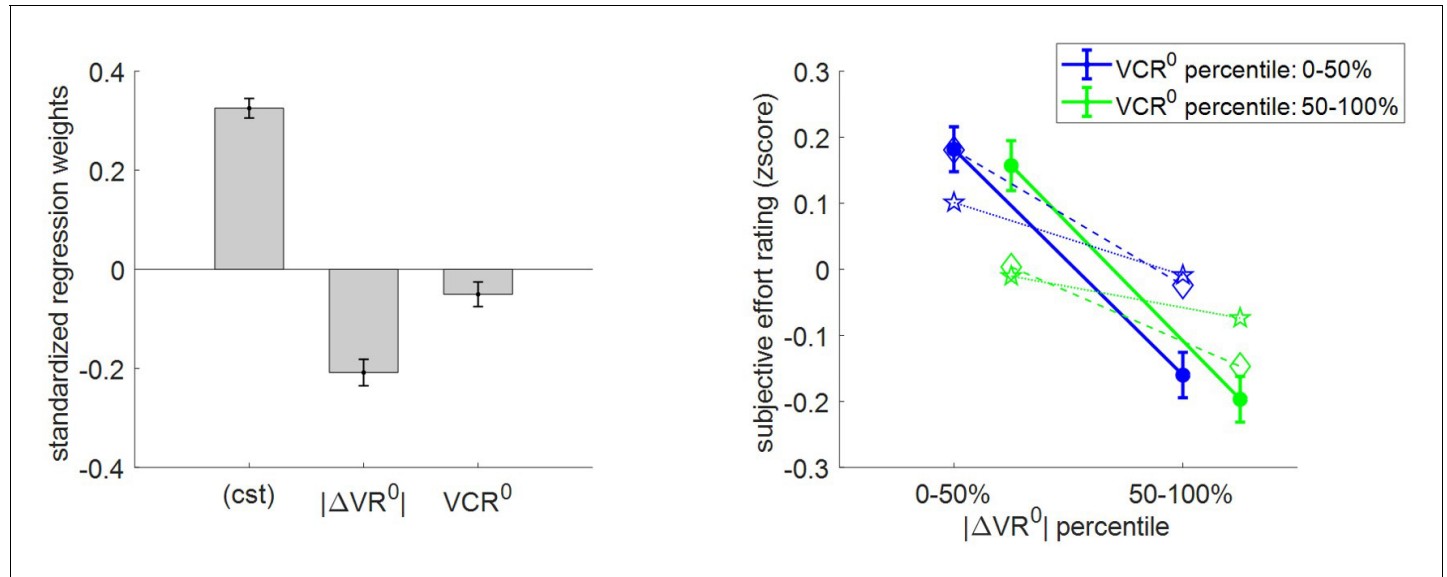

**Figure 6.** Three-way relationship between subjective effort rating, value, and value certainty. Same format as *Figure 5*.

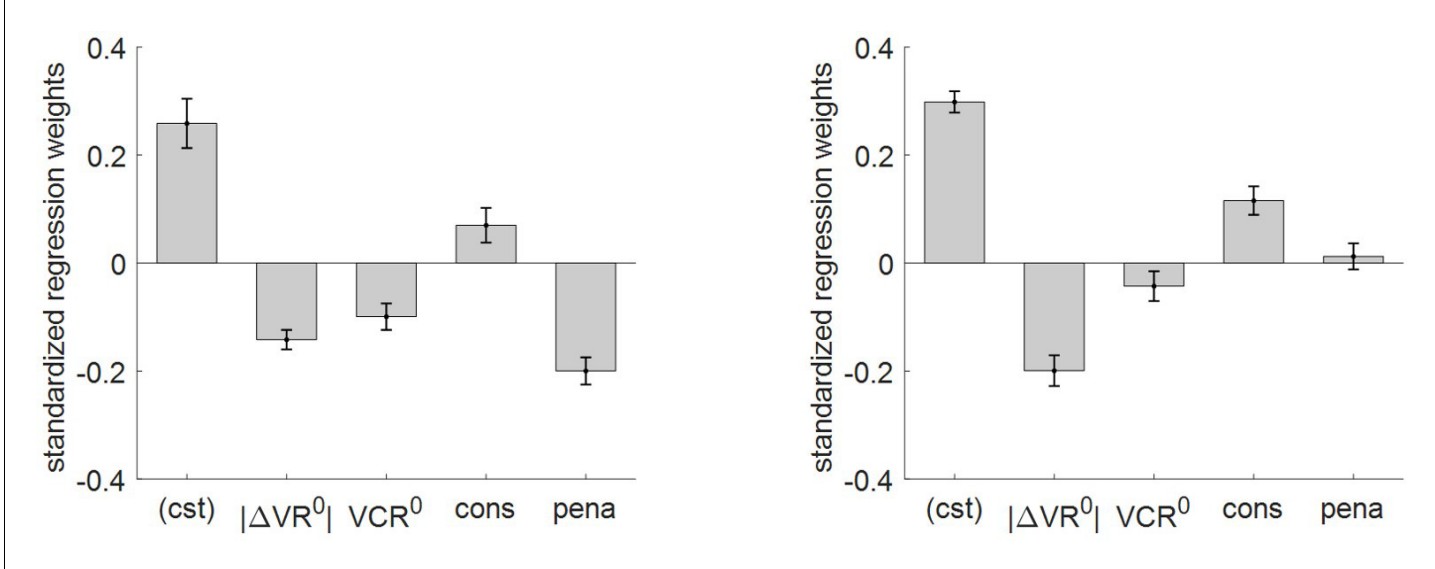

**Figure 7.** Impact of consequential and penalized conditions on effort-related variables. Left panel: log-RT: mean standardized regression weights (same format as *Figure 4* – left panel, *cons* = 'consequential' condition, *pena* = 'penalized' condition). Right panel: subjective effort ratings: same format as left panel.

p=0.036; effort ratings: mean standardized regression weight = 0.12, s.e.m. = 0.03, $p<10^{-3}$; one-sided t-tests). In addition, response times are significantly faster for 'penalized' than for 'neutral' decisions (mean standardized regression weight = $-0.26$, s.e.m. = 0.03, $p<10^{-3}$; one-sided t-test). However, the difference in subjective effort ratings between 'neutral' and 'penalized' decisions does not reach statistical significance (mean effort difference = 0.012, s.e.m. = 0.024, p=0.66; two-sided t-test). We will comment on this in the Discussion section.

## Predicting decision-related variables

Under the MCD model, 'decision-related' dependent variables (i.e., choice confidence, change of mind, spreading of alternatives, and value certainty gain) are determined by the amount of resources allocated to the decision. However, their relationship to features of prior value representation is not trivial (see section 2 of the Appendix for the specific case of choice confidence). For this reason, we will recapitulate the qualitative MCD prediction that can be made about each of them, prior to summarizing the empirical data and its corresponding postdictions and out-of-sample predictions. Note that here, the latter are obtained from a model fit on effort-related variables only.

First, we checked how choice confidence relates to $|\Delta VR^0|$ and $VCR^0$. Under the MCD model, choice confidence reflects the discriminability of the options' value representations after optimal resource allocation. Recall that more resources are allocated to the decision when either $|\Delta VR^0|$ or $VCR^0$ decreases. However, under moderate effort efficacies, this does not overcompensate decision difficulty, and thus choice confidence should decrease. As with effort-related variables, we regressed trial-by-trial confidence data against $|\Delta VR^0|$ and $VCR^0$, and then performed a group-level random-effect analysis on regression weights. The results of this analysis, as well as the comparison between empirical, predicted, and postdicted confidence data are shown in *Figure 8*.

The results of the group-level random effect analysis confirm our qualitative predictions. In brief, both $|\Delta VR^0|$ (mean standardized regression weight = 0.25, s.e.m. = 0.02, $p<10^{-3}$; one-sided t-test) and $VCR^0$ (mean standardized regression weight = 0.16, s.e.m. = 0.03, $p<10^{-3}$; one-sided t-test) have a significant positive impact on choice confidence. Here again, MCD postdictions and out-of-sample predictions are remarkably accurate at capturing the effect of both $|\Delta VR^0|$ and $VCR^0$ variables (though predictions slightly underestimate the effect of $|\Delta VR^0|$).

Second, we checked how change of mind relates to $|\Delta VR^0|$ and $VCR^0$. Note that we define a change of mind according to two criteria: (i) the choice is incongruent with the prior preference inferred from the pre-choice value ratings, and (ii) the choice is congruent with the posterior

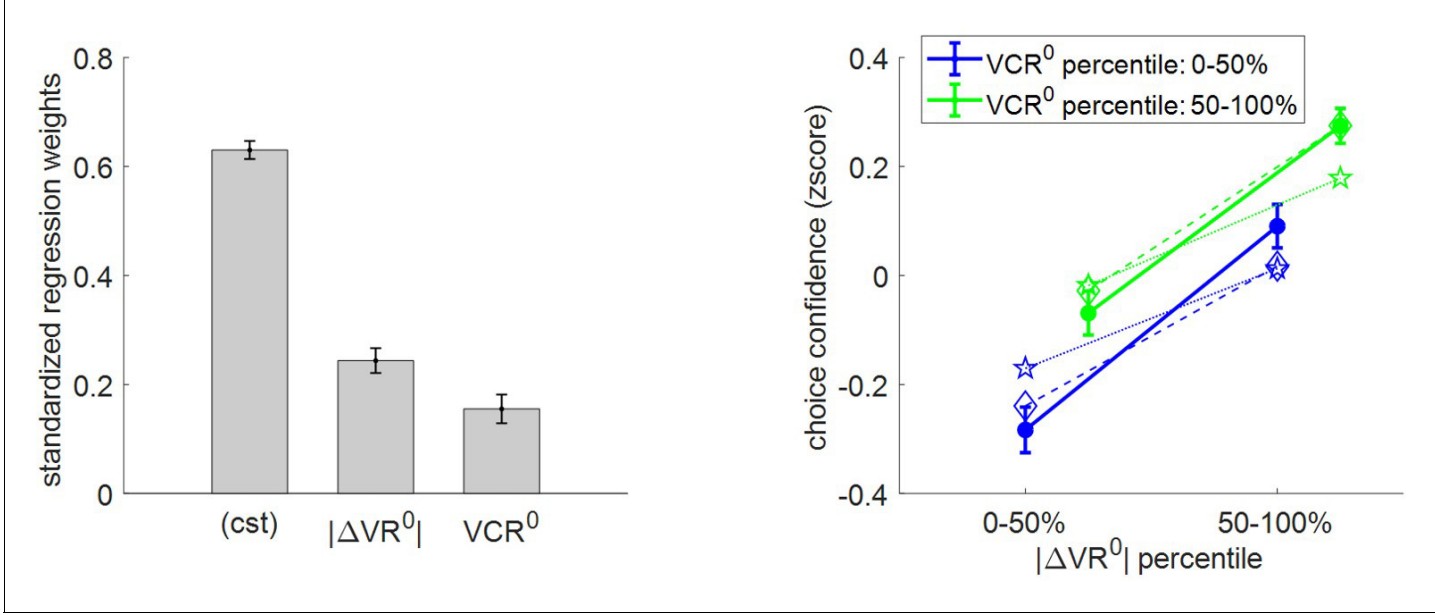

**Figure 8.** Three-way relationship between choice confidence, value, and value certainty. Same format as *Figure 5*.

preference inferred from post-choice value ratings. The latter criterion distinguishes a change of mind from a mere 'error', which may arise from attentional and/or motor lapses. Under the MCD model, we expect no change of mind unless type #2 efficacy $\gamma \neq 0$. In addition, the rate of change of mind should decrease when either $|\Delta VR^0|$ or $VCR^0$ increases. This is because increasing $|\Delta VR^0|$ and/or $VCR^0$ will decrease the demand for effort, which implies that the probability of reversing the prior preference will be smaller. *Figure 9* shows the corresponding model predictions/postdictions and summarizes the corresponding empirical data.

Note that, on average, the rate of change of mind reaches about 14.5% (s.e.m. = 0.008, $p < 10^{-3}$, one-sided t-test), which is significantly higher than the rate of 'error' (mean rate difference = 2.3%, s. e.m. = 0.01, p=0.032; two-sided t-test). The results of the group-level random effect analysis confirm

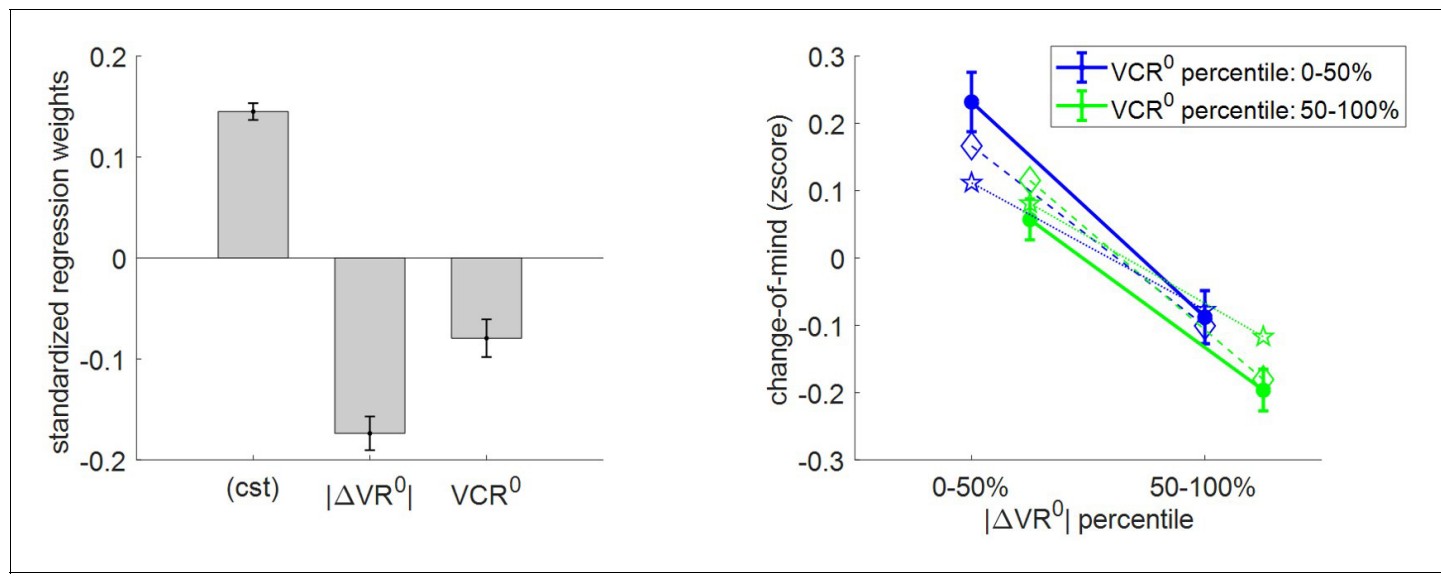

**Figure 9.** Three-way relationship between change of mind, value, and value certainty. Same format as *Figure 5*.

our qualitative MCD predictions. In brief, both $|\Delta VR^0|$ (mean standardized regression weight = $-0.17$, s.e.m. = 0.02, p<$10^{-3}$; one-sided t-test) and $VCR^0$ (mean standardized regression weight = $-0.08$, s.e.m. = 0.03, p<$10^{-3}$; one-sided t-test) have a significant negative impact on the rate of change of mind. Again, MCD postdictions and out-of-sample predictions are remarkably accurate at capturing the effect of both $|\Delta VR^0|$ and $VCR^0$ variables (though predictions slightly underestimate the effect of $|\Delta VR^0|$).

Third, we checked how spreading of alternatives relates to $|\Delta VR^0|$ and $VCR^0$. Recall that spreading of alternatives measures the magnitude of choice-induced preference change. Under the MCD model, the reported value of alternative options cannot spread apart unless type #2 efficacy $\gamma \neq 0$. In addition, and as with change of mind, spreading of alternatives should globally follow the optimal effort allocation, that is, it should decrease when $|\Delta VR^0|$ and/or $VCR^0$ increase. *Figure 10* shows the corresponding model predictions/postdictions and summarizes the corresponding empirical data.

One can see that there is a significant positive spreading of alternatives (mean = 0.04 A.U., s.e.m. = 0.004, p<$10^{-3}$, one-sided t-test). This is reassuring, because it dismisses the possibility that $\gamma = 0$ (which would mean that effort does not perturb the mode of value representations). In addition, the results of the group-level random effect analysis confirm that both $|\Delta VR^0|$ (mean standardized regression weight = $-0.09$, s.e.m. = 0.03, p=0.001; one-sided t-test) and $VCR^0$ (mean standardized regression weight = $-0.04$, s.e.m. = 0.02, p=0.03; one-sided t-test) have a significant negative impact on spreading of alternatives. Note that this replicates previous findings on choice-induced preference change (*Lee and Coricelli, 2020*; *Lee and Daunizeau, 2020*). Finally, MCD postdictions and out-of-sample predictions accurately capture the effect of both $|\Delta VR^0|$ and $VCR^0$ variables in a quantitative manner (though predictions slightly underestimate the effect of $|\Delta VR^0|$).

Fourth, we checked how $|\Delta VR^0|$ and $VCR^0$ impact value certainty gain. Under the MCD model, the certainty of value representations cannot improve unless type #1 efficacy $\beta \neq 0$. In addition, value certainty gain should globally follow the optimal effort allocation, i.e., it should decrease when $|\Delta VR^0|$ and/or $VCR^0$ increase. *Figure 11* shows the corresponding model predictions/postdictions and summarizes the corresponding empirical data.

Importantly, there is a small but significantly positive certainty gain (mean = 0.11 A.U., s.e.m. = 0.06, p=0.027, one-sided t-test). This is reassuring, because it dismisses the possibility that $\beta = 0$ (which would mean that effort does not increase the precision of value representation). This time, the results of the group-level random effect analysis only partially confirm our qualitative MCD predictions. In brief, although $VCR^0$ has a very strong negative impact on certainty gain (mean standardized regression weight = $-0.61$, s.e.m. = 0.04, p<$10^{-3}$; one-sided t-test), the effect of $|\Delta VR^0|$ does not reach statistical significance (mean standardized regression weight = $-0.009$, s.e.m. = 0.01, p=0.35; one-sided t-test). We note that a simple regression-to-the-mean artifact

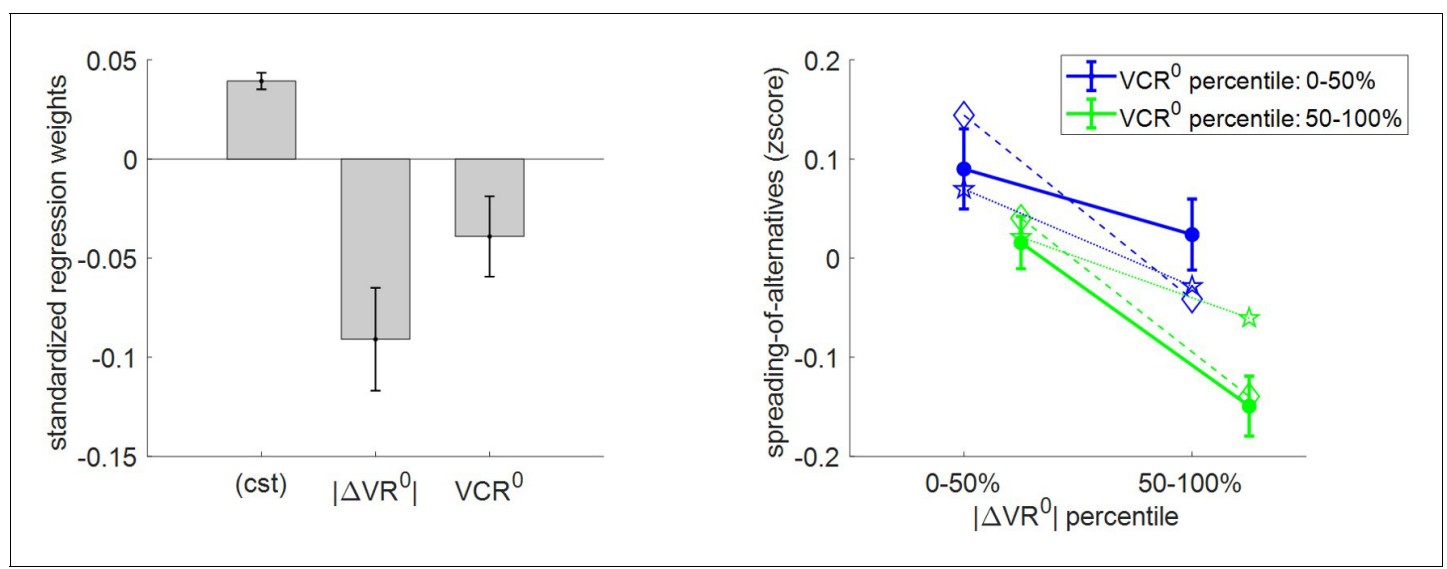

**Figure 10.** Three-way relationship between spreading of alternatives, value, and value certainty. Same format as *Figure 5*.

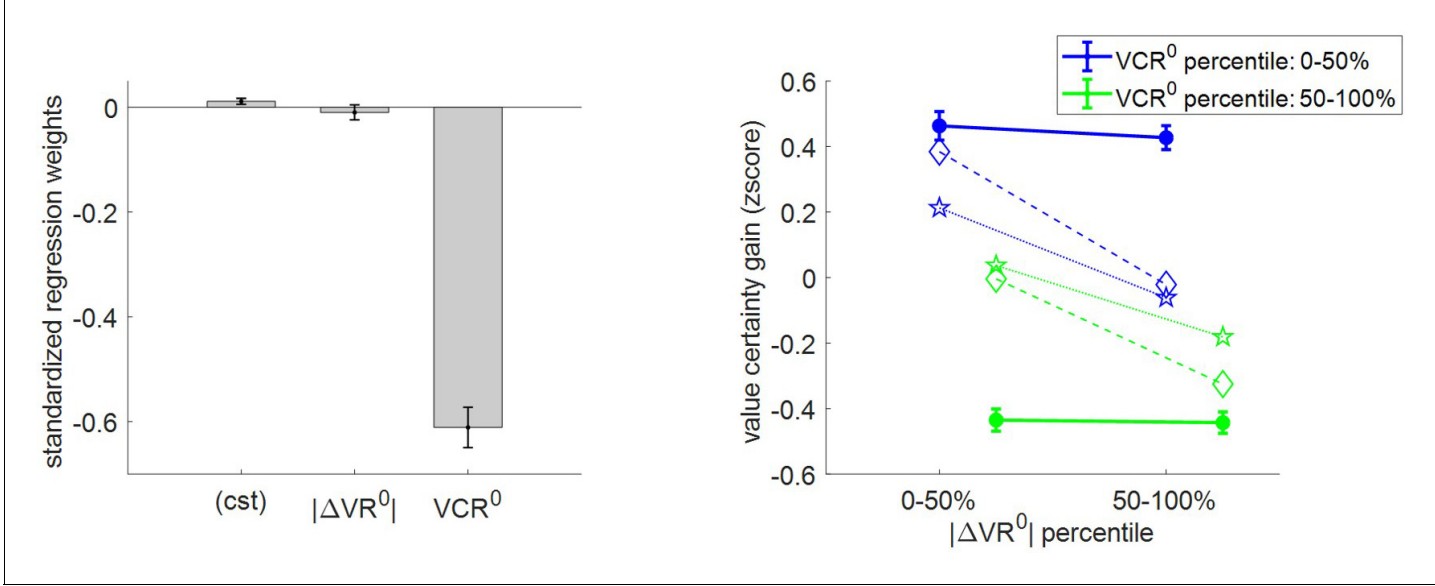

**Figure 11.** Three-way relationship between value certainty gain, value, and value certainty. Same format as *Figure 5*.

(*Stigler, 1997*) likely inflates the observed negative correlation between $VCR^0$ and certainty gain, beyond what would be predicted under the MCD model. Accordingly, both MCD postdictions and out-of-sample predictions clearly underestimate the effect of $VCR^0$ (and overestimate the effect of $|\Delta VR^0|$).

## Discussion

In this work, we have presented a novel computational model of decision-making that explains the intricate relationships between effort-related variables (response time, subjective effort) and decision-related variables (choice confidence, change of mind, spreading of alternatives, and choice-induced value certainty gain). This model assumes that deciding between alternative options whose values are uncertain induces a demand for allocating cognitive resources to value-relevant information processing. Cognitive resource allocation then optimally trades mental effort for confidence, given the prior discriminability of the value representations.

Such metacognitive control of decisions or MCD provides an alternative theoretical framework to accumulation-to-bound models of decision-making, e.g., drift-diffusion models or DDMs (*Milosavljevic et al., 2010*; *Ratcliff et al., 2016*; *Tajima et al., 2016*). Recall that DDMs assume that decisions are triggered once the noisy evidence in favor of a particular option has reached a predefined bound. Standard DDM variants make quantitative predictions regarding both response times and decision outcomes, but are agnostic about choice confidence, spreading of alternatives, value certainty gain, and/or subjective effort ratings. We note that simple DDMs are significantly less accurate than MCD at making out-of-sample predictions on dependent variables common to both models (e.g., change of mind). We refer the reader interested in the details of the MCD–DDM comparison to section 9 of the Appendix.

But how do MCD and accumulation-to-bound models really differ? For example, if the DDM can be understood as an optimal policy for value-based decision-making (*Tajima et al., 2016*), then how can these two frameworks both be optimal? The answer lies in the distinct computational problems that they solve. The MCD solves the problem of finding the optimal amount of effort to invest under the possibility that yet-unprocessed value-relevant information might change the decision maker's mind. In fact, this resource allocation problem would be vacuous, would it not be possible to reassess preferences during the decision process. In contrast, the DDM provides an optimal solution to the problem of efficiently comparing option values, which may be unreliably signaled, but remain nonetheless stationary. Of course, the DDM decision variable (i.e., the 'evidence' for a given choice option over the alternative) may fluctuate, e.g. it may first drift toward the upper bound, but then

eventually reach the lower bound. This is the typical DDM's explanation for why people change their mind over the course of deliberation (*Kiani et al., 2014*; *Resulaj et al., 2009*). But, critically, these fluctuations are not caused by changes in the underlying value signal (i.e., the DDM's drift term). Rather, the fluctuations are driven by neural noise that corrupts the value signals (i.e., the DDM's diffusion term). This is why the DDM cannot predict choice-induced preference changes, or changes in options' values more generally. This distinction between MCD and DDM extends to other types of accumulation-to-bound models, including race models (*De Martino et al., 2013*; *Tajima et al., 2019*). We note that either of these models (DDM or race) could be equipped with pre-choice value priors (initial bias), and then driven with 'true' values (drift term) derived from post-choice ratings. But then, simulating these models would require both pre-choice and post-choice ratings, which implies that choice-induced preference changes cannot be *predicted* from pre-choice ratings using a DDM. In contrast, the MCD model assumes that the value representations themselves are modified during the decision process, in proportion to the effort expenditure. Now the latter is maximal when prior value difference is minimal, at least when type #2 efficacy dominates (γ-effect, see section 2 of the Appendix). In turn, the MCD model predicts that the magnitude of (choice-induced) value spreading should decrease when the prior value difference increases (cf. *Equation 14*). Together with (choice-induced) value certainty gain, this quantitative prediction is unique to the MCD framework, and cannot be derived from existing variants of DDM.

As a side note, the cognitive essence of spreading of alternatives has been debated for decades. Its typical interpretation is that of 'cognitive dissonance' reduction: if people feel uneasy about their choice, they later convince themselves that the chosen (rejected) item was actually better (worse) than they originally thought (*Bem, 1967*; *Harmon-Jones et al., 2009*; *Izuma and Murayama, 2013*). In contrast, the MCD framework would rather suggest that people tend to reassess value representations until they reach a satisfactory level of confidence prior to committing to their choice. Interestingly, recent neuroimaging studies have shown that spreading of alternatives can be predicted from brain activity measured during the decision (*Colosio et al., 2017*; *Jarcho et al., 2011*; *Kitayama et al., 2013*; *van Veen et al., 2009*, *Voigt et al., 2019*). This is evidence against the idea that spreading of alternatives only occurs after the choice has been made. In addition, key regions of the brain's valuation and cognitive control systems are involved, including: the right inferior frontal gyrus, the ventral striatum, the anterior insula, and the anterior cingulate cortex (ACC). This further corroborates the MCD interpretation, under the assumption that the ACC is involved in controlling the allocation of cognitive effort (*Musslick et al., 2015*; *Shenhav et al., 2013*). Having said this, both MCD and cognitive dissonance reduction mechanisms may contribute to spreading of alternatives, on top of its known statistical artifact component (*Chen and Risen, 2010*). The latter is a consequence of the fact that pre-choice value ratings may be unreliable and is known to produce an apparent spreading of alternatives that decreases with pre-choice value difference (*Izuma and Murayama, 2013*). Although this pattern is compatible with our results, the underlying statistical confound is unlikely to drive our results. The reason is twofold. First, effort-related variables yield accurate within-subject out-of-sample predictions about spreading of alternatives (cf. *Figure 10*). Second, we have already shown that the effect of pre-choice value difference on spreading of alternatives is higher here than in a control condition where the choice is made after both rating sessions (*Lee and Daunizeau, 2020*).

A central tenet of the MCD model is that involving cognitive resources in value-related information processing is costly, which calls for an efficient resource allocation mechanism. A related notion is that information processing resources may be limited, in particular: value-encoding neurons may have a bounded firing range (*Louie and Glimcher, 2012*). In turn, 'efficient coding' theory assumes that the brain has evolved adaptive neural codes that optimally account for such capacity limitations (*Barlow, 1961*; *Laughlin, 1981*). In our context, efficient coding implies that value-encoding neurons should optimally adapt their firing range to the prior history of experienced values (*Polanía et al., 2019*). When augmented with a Bayesian model of neural encoding/decoding (*Wei and Stocker, 2015*), this idea was successful in explaining the non-trivial relationship between choice consistency and the distribution of subjective value ratings. Both MCD and efficient coding frameworks assume that value representations are uncertain, which stresses the importance of metacognitive processes in decision-making control (*Fleming and Daw, 2017*). However, they differ in how they operationalize the notion of efficiency. In efficient coding, the system is 'efficient' in the sense that it changes the physiological properties of value-encoding neurons to minimize the information loss that results

from their limited firing range. In MCD, the system is 'efficient' in the sense that it allocates the amount of resources that optimally trades effort cost against choice confidence. These two perspectives may not be easy to reconcile. A possibility is to consider, for example, energy-efficient population codes (*Hiratani and Latham, 2020*; *Yu et al., 2016*), which would tune the amount of neural resources involved in representing value to optimally trade information loss against energetic costs.

Now, let us highlight that the MCD model offers a plausible alternative interpretation for the two main reported neuroimaging findings regarding confidence in value-based choices (*De Martino et al., 2013*). First, the ventromedial prefrontal cortex or vmPFC was found to respond positively to both value difference (i.e., $\Delta VR^0$) and choice confidence. Second, the right rostrolateral prefrontal cortex or rRLPFC was more active during low-confidence versus high-confidence choices. These findings were originally interpreted through a so-called 'race model', in which a decision is triggered whenever the first of option-specific value accumulators reaches a bound. Under this model, choice confidence is defined as the final gap between the two value accumulators. We note that this scenario predicts the same three-way relationship between response time, choice outcome, and choice confidence as the MCD model (see section 7 of the Appendix). In brief, rRLPFC was thought to perform a readout of choice confidence (for the purpose of subjective metacognitive report) from the racing value accumulators hosted in the vmPFC. Under the MCD framework, the contribution of the vmPFC to value-based decision control might rather be to construct item values, and to anticipate and monitor the benefit of effort investment (i.e., confidence). This would be consistent with recent fMRI studies suggesting that vmPFC confidence computations signal the attainment of task goals (*Hebscher and Gilboa, 2016*; *Lebreton et al., 2015*). Now, recall that the MCD model predicts that confidence and effort should be anti-correlated. Thus, the puzzling negative correlation between choice confidence and rRLPFC activity could be simply explained under the assumption that rRLPFC provides the neurocognitive resources that are instrumental for processing value-relevant information during decisions (and/or to compare item values). This resonates with the known involvement of rRLPFC in reasoning (*Desrochers et al., 2015*; *Dumontheil, 2014*) or memory retrieval (*Benoit et al., 2012*; *Westphal et al., 2019*).

At this point, we note that the current MCD model clearly has limited predictive power. Arguably, this limitation is partly due to the imperfect reliability of the data, and to the fact that MCD does not model all decision-relevant processes. In addition, assigning variations in many effort- and/or decision-related variables to a unique mechanism with few degrees of freedom necessarily restricts the model's expected predictive power. Nevertheless, the MCD model may also not yield a sufficiently tight approximation to the mechanism that it focuses on. In turn, it may unavoidably distort the impact of prior value representations and other decision input variables. The fact that it can only explain 81% of the variability in dependent variables that can be captured using simple linear regressions against ΔVR0 and VCR0 (see section 11 of the Appendix) supports this notion. A likely explanation here is that the MCD model includes constraints that prevent it from matching the model-free postdiction accuracy level. In turn, one may want to extend the MCD model with the aim of relaxing these constraints. For example, one may allow for deviations from the optimal resource allocation framework, e.g., by considering candidate systematic biases whose magnitudes would be controlled by specific additional parameters. Having said this, some of these constraints may be necessary, in the sense that they derive from the modeling assumptions that enable the MCD model to provide a unified explanation for all dependent variables (and thus make out-of-sample predictions). What follows is a discussion of what we perceive as the main limitations of the current MCD model, and the directions of improvement they suggest.

First, we did not specify what determines decision 'importance', which effectively acts as a weight for confidence against effort costs (cf. *R* in *Equation 2* of the Model section). We know from the comparison of 'consequential' and 'neutral' choices that increasing decision importance eventually increases effort, as predicted by the MCD model. However, decision importance may have many determinants, such as, for example, the commitment duration of the decision (e.g., life partner choices), the breadth of its repercussions (e.g., political decisions), or its instrumentality with respect to the achievement of superordinate goals (e.g., moral decisions). How these determinants are combined and/or moderated by the decision context is virtually unknown (*Locke and Latham, 2002*; *Locke and Latham, 2006*). In addition, decision importance may also be influenced by the prior (intuitive/emotional/habitual) appraisal of choice options. For example, we found that, all else equal, people spent more time and effort deciding between two disliked items than between two liked

items (results not shown). This reproduces recent results regarding the evaluation of choice sets (*Shenhav and Karmarkar, 2019*). One may also argue that people should care less about decisions between items that have similar values (*Oud et al., 2016*). In other terms, decision importance would be an increasing function of the absolute difference in pre-choice value ratings. However, this would predict that people invest fewer resources when deciding between items of similar pre-choice values, which directly contradicts our results (cf. *Figures 5* and *6*). Importantly, options with similar values may still be very different from each other, when decomposed on some value-relevant feature space. For example, although two food items may be similarly liked and/or wanted, they may be very different in terms of, e.g., tastiness and healthiness, which would induce some form of decision conflict (*Hare et al., 2009*). In such a context, making a decision effectively implies committing to a preference about feature dimensions. This may be deemed to be consequential, when contrasted with choices between items that are similar in all regards. In turn, decision importance may rather be a function of options' feature conflict. In principle, this alternative possibility is compatible with our results, under the assumption that options' feature conflict is approximately orthogonal to pre-choice value difference. Considering how decision importance varies with feature conflict may significantly improve the amount of explained trial-by-trial variability in the model's dependent variables. We note that the brain's quick/automatic assessment of option features may also be the main determinant of the prior value representations that eventually serve to compute the MCD-optimal resource allocation. Probing these computational assumptions will be the focus of forthcoming publications.

Second, our current version of the MCD model relies on a simple variant of resource costs and efficacies. One may thus wonder how sensitive model predictions are to these assumptions. For example, one may expect that type #2 efficacy saturates, i.e. that the magnitude of the perturbation $\delta(z)$ to the modes $\mu(z)$ of the value representations eventually reaches a plateau instead of growing linearly with $z$ (cf. *Equation 6*). We thus implemented and tested such a model variant. We report the results of this analysis in section 10 of the Appendix. In brief, a saturating type #2 efficacy brings no additional explanatory power for the model's dependent variables. Similarly, rendering the cost term nonlinear (e.g., quadratic) does not change the qualitative nature of the MCD predictions. More problematic, perhaps, is the fact that we did not consider distinct types of effort, which could, in principle, be associated with different costs and/or efficacies. For example, the efficacy of allocating attention may depend upon which option is considered. In turn, the brain may dynamically refocus its attention on maximally uncertain options when prospective information gains exceed switch costs (*Callaway et al., 2021*; *Jang et al., 2021*). Such optimal adjustment of divided attention might eventually explain systematic decision biases and shortened response times for 'default' choices (*Lopez-Persem et al., 2016*). Another possibility is that effort might be optimized along two canonical dimensions, namely duration and intensity. The former dimension essentially justifies the fact that we used RT as a proxy for the amount of allocated resources. This is because, if effort intensity stays constant, then longer RT essentially signals greater resource expenditure. In fact, as is evident from the comparison between 'penalized' and 'neutral' choices, imposing an external penalty cost on RT reduces, as expected, the ensuing effort duration. More generally, however, the dual optimization of effort dimensions might render the relationship between effort and RT more complex. For example, beyond memory span or attentional load, effort intensity could be related to processing speed. This would explain why, although 'penalized' choices are made much faster than 'neutral' choices, the associated subjective feeling of effort is not as strongly impacted as RT (cf. *Figure 7*). In any case, the relationship between effort and RT might depend upon the relative costs and/or efficacies of effort duration and intensity, which might themselves be partially driven by external availability constraints (cf. time pressure or multitasking). We note that the essential nature of the cost of mental effort in cognitive tasks (e.g., neurophysiological cost, interferences cost, or opportunity cost) is still a matter of intense debate (*Kurzban et al., 2013*; *Musslick et al., 2015*; *Ozcimder et al., 2017*). Progress toward addressing this issue will be highly relevant for future extensions of the MCD model.

Third, we did not consider the issue of identifying plausible neuro-computational implementations of MCD. This issue is tightly linked to the previous one, in that distinct cost types would likely impose different constraints on candidate neural network architectures (*Feng et al., 2014*; *Petri et al., 2017*). For example, underlying brain circuits are likely to operate MCD in a more reactive manner, eventually adjusting resource allocation from the continuous monitoring of relevant

decision variables (e.g., experienced costs and benefits). Such a reactive process contrasts with our current, prospective-only variant of MCD, which sets resource allocation based on anticipated costs and benefits. We already checked that simple reactive scenarios, where the decision is triggered whenever the online monitoring of effort or confidence reaches the optimal threshold, make predictions qualitatively similar to those we have presented here. We tend to think, however, that such reactive processes should be based on a dynamic programming perspective on MCD, as was already done for the problem of optimal efficient value comparison (*Tajima et al., 2016*; *Tajima et al., 2019*). We will pursue this and related neuro-computational issues in subsequent publications.

## Code availability

The computer code and algorithms that support the findings of this study will soon be made available from the open academic freeware VBA (http://mbb-team.github.io/VBA-toolbox/). Until then, they are available from the corresponding author upon reasonable request.

## Ethical compliance

This study complies with all relevant ethical regulations and received formal approval from the INSERM Ethics Committee (CEEI-IRB00003888, decision no 16–333). In particular, in accordance with the Helsinki declaration, all participants gave written informed consent prior to commencing the experiment, which included consent to disseminate the results of the study via publication.

# Acknowledgements

DL was supported by a grant from the Laboratory of Excellence of Biology for Psychiatry (LabEx BIO-PSY, Paris, France).

# Additional information

### Funding

| Funder | Author |
| --- | --- |
| LabEx BIOPSY | Douglas G Lee |

The funders had no role in study design, data collection and interpretation, or the decision to submit the work for publication.

### Author contributions

Douglas G Lee, Conceptualization, Resources, Data curation, Software, Formal analysis, Funding acquisition, Validation, Investigation, Visualization, Methodology, Writing - original draft, Project administration, Writing - review and editing; Jean Daunizeau, Conceptualization, Resources, Software, Formal analysis, Supervision, Funding acquisition, Validation, Visualization, Methodology, Writing - original draft, Project administration, Writing - review and editing

### Author ORCIDs

Douglas G Lee (iD) https://orcid.org/0000-0001-5892-8694
Jean Daunizeau (iD) https://orcid.org/0000-0001-9142-1270

### Ethics

Human subjects: This study complies with all relevant ethical regulations and received formal approval from the INSERM Ethics Committee (CEEI-IRB00003888, decision no 16-333). In particular, in accordance with the Helsinki declaration, all participants gave written informed consent prior to commencing the experiment, which included consent to disseminate the results of the study via publication.

### Decision letter and Author response

Decision letter https://doi.org/10.7554/eLife.63282.sa1

Author response https://doi.org/10.7554/eLife.63282.sa2

# Additional files

## Supplementary files
- Source data 1. Behavioral data.
- Source code 1. Analysis code.
- Transparent reporting form

## Data availability

Empirical data as well as model fitting code have been uploaded as part of this submission. Also, it is now publicly available at Dryad: https://doi.org/10.5061/dryad.7h44j0zsg.

The following dataset was generated:

| Author(s) | Year | Dataset title | Dataset URL | Database and Identifier |
|---|---|---|---|---|
| Lee D, Daunizeau J | 2021 | Lee and Daunizeau choice data from: Trading mental effort for confidence in the metacognitive control of value-based decision-making | https://doi.org/10.5061/dryad.7h44j0zsg | Dryad Digital Repository, 10.5061/dryad.7h44j0zsg |

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

## Appendix 1

### 1. On the approximation accuracy of the expected confidence gain

The MCD model relies on the system's ability to anticipate the benefit of allocating resources to the decision process. Given the mathematical expression of choice confidence (*Equation 4* in the main text), this reduces to finding an analytical approximation to the following expression:

$$\bar{P} = E[s(\lambda|x|)] \tag{A1}$$

where $x \to s(x) = 1/1 + e^{-x}$ is the sigmoid mapping, $\lambda$ is an arbitrary constant, and the expectation is taken under the Gaussian distribution of $x \sim N(\mu, \sigma^2)$, whose mean and variance are $\mu$ and $\sigma^2$, respectively.

Note that the absolute value mapping $x \to |x|$ follows a folded normal distribution, whose first two moments $E[|x|]$ and $V[|x|]$ have known expressions:

$$\begin{cases} E[|x|] = \sigma\sqrt{\dfrac{2}{\pi}}exp\left(-\dfrac{|\mu|^2}{2\sigma^2}\right) + \mu\left(2 \times s\left(\dfrac{\pi\mu}{\sigma\sqrt{3}}\right) - 1\right) \\ V[|x|] = \mu^2 + \sigma^2 - E[|x|]^2 \end{cases} \tag{A2}$$

where the first line relies on a moment-matching approximation to the cumulative normal distribution function (*Daunizeau, 2017a*). This allows us to derive the following analytical approximation to *Equation A1*:

$$\bar{P} \approx s\left(\frac{E[|x|]}{\sqrt{\frac{1}{\lambda^2} + aV[|x|]}}\right) \tag{A3}$$

where setting $a \approx 3/\pi^2$ makes this approximation tight (*Daunizeau, 2017a*).

The quality of this approximation can be evaluated by drawing samples of $x \sim N(\mu, \sigma^2)$, and comparing the Monte-Carlo average of $s(\lambda|x|)$ with the expression given in *Equation A3*. This is summarized in *Appendix 1—figure 1*, where the range of variation for the moments of $x$ was set as follows: $\mu \in [-4, 4]$ and $\sigma^2 \in [0, 4]$.

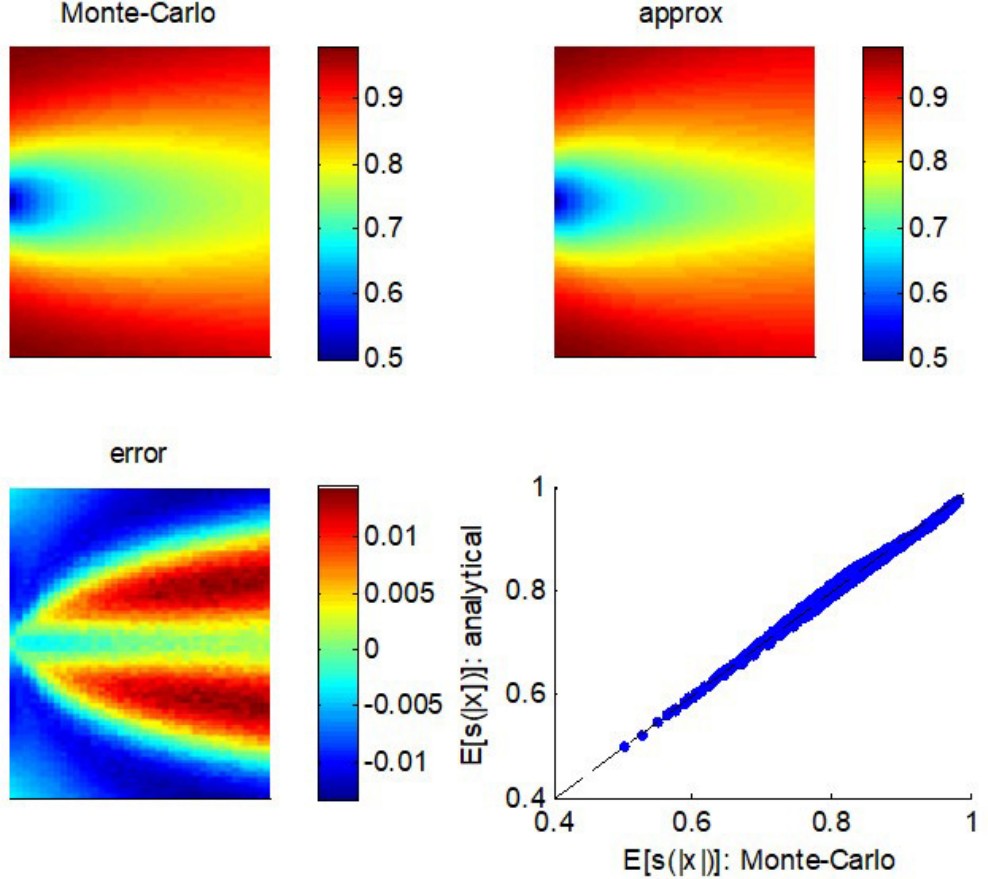

**Appendix 1—figure 1.** Quality of the analytical approximation to $\bar{P}$. Upper left panel: the Monte-Carlo estimate of $\bar{P}$ (color-coded) is shown as a function of both the mean $\mu \in [-4, 4]$ (y-axis) and the variance $\sigma^2 \in [0, 4]$ (x-axis) of the parent process $x \sim N(\mu, \sigma^2)$. Upper right panel: analytic approximation to $\bar{P}$ as given by *Equation A3* (same format). Lower left panel: the error, that is, the difference between the Monte-Carlo and the analytic approximation (same format). Lower right panel: the analytic approximation (y-axis) is plotted as a function of the Monte-Carlo estimate (x-axis) for each pair of moments $\{\mu, \sigma^2\}$ of the parent distribution.

One can see that the error rarely exceeds 5%, across the whole range of moments $\{\mu, \sigma^2\}$ of the parent distribution. This is how tight the analytic approximation of the expected confidence gain (*Equation 9* in the main text) is.

## 2. On the impact of model parameters for the MCD model

To begin with, note that the properties of the metacognitive control of decisions (in terms of effort allocation and/or confidence) actually depend on the demand for resources, which is itself determined by prior value representations (or, more properly, by the prior uncertainty $\sigma^0$ and the absolute means' difference $|\Delta\mu^0|$). Now, the way the MCD-optimal control responds to the resource demand is determined by effort efficacy and unitary cost parameters. In addition, MCD-optimal confidence may not trivially follow resource allocation, because it may be overcompensated by choice difficulty.

First, recall that the amount $\hat{z}$ of allocated resources maximizes the EVC:

$$\hat{z} = \arg\max_z [R \times \bar{P}_c(z) - \alpha z] \tag{A4}$$

where $\bar{P}_c(z)$ is given in *Equation 9* in the main text. According to the implicit function theorem, the derivatives of $\hat{z}$ w.r.t. $\sigma^0$ and $|\Delta\mu^0|$ are given by *Gould et al., 2016*:

$$
\begin{cases}
\dfrac{\partial \hat{z}}{\partial |\Delta\mu^0|} = -\dfrac{\left.\dfrac{\partial^2 \bar{P}_c(z)}{\partial |\Delta\mu^0|\partial z}\right|_{z=\hat{z}}}{\left.\dfrac{\partial^2 \bar{P}_c(z)}{\partial |\Delta\mu^0|^2}\right|_{z=\hat{z}}} \\[3em]
\dfrac{\partial \hat{z}}{\partial \sigma^0} = -\dfrac{\left.\dfrac{\partial^2 \bar{P}_c(z)}{\partial \sigma^0\partial z}\right|_{z=\hat{z}}}{\left.\dfrac{\partial^2 \bar{P}_c(z)}{\partial \sigma^{02}}\right|_{z=\hat{z}}}
\end{cases}
\tag{A5}
$$

The double derivatives in *Equations A5* are not trivial to obtain.

First, the gradient $\partial \bar{P}_c(z)/\partial |\Delta\mu^0|$ of choice confidence w.r.t. $|\Delta\mu^0|$ writes:

$$
\begin{aligned}
\frac{\partial \bar{P}_c(z)}{\partial |\Delta\mu^0|} &= \frac{\partial \bar{P}_c(z)}{\partial E[|\Delta\mu||z]}\frac{\partial E[|\Delta\mu||z]}{\partial |\Delta\mu^0|} + \frac{\partial \bar{P}_c(z)}{\partial V[|\Delta\mu||z]}\frac{\partial V[|\Delta\mu||z]}{\partial |\Delta\mu^0|} \\
&= 3K(z)\left(\left(2\sigma(z) + 2\gamma z + |\Delta\mu^0|^2\right)\frac{\partial E[|\Delta\mu||z]}{\partial |\Delta\mu^0|} - |\Delta\mu^0|E[|\Delta\mu||z]\right)
\end{aligned}
\tag{A6}
$$

where $K(z) \geq 0$ is given by:

$$
K(z) = \frac{\pi \bar{P}_c(z)(1 - \bar{P}_c(z))}{(6\sigma(z) + 3V[|\Delta\mu||z])^{\frac{3}{2}}}
\tag{A7}
$$

Note that the gradient $\partial E[|\Delta\mu||z]/\partial |\Delta\mu^0| \geq 0$ in *Equation A6* can be obtained analytically from *Equation 7* in the main text. However, we refrain from doing this, because it is clear that deriving the right-hand term of *Equation A6* w.r.t. both $\sigma^0$ and $z$ will not bring any simple insight regarding the impact of $|\Delta\mu^0|$ onto $\hat{z}$.

Also, although the gradient $\partial \bar{P}_c(\hat{z})/\partial \sigma^0$ of choice confidence wr.t. $\sigma^0$ takes a much more concise form:

$$
\begin{aligned}
\frac{\partial \bar{P}_c(z)}{\partial \sigma^0} &= \frac{\partial \bar{P}_c(z)}{\partial \sigma(z)}\frac{\partial \sigma(z)}{\partial \sigma^0} \\
&= -\frac{3K(z)E[|\Delta\mu||z]}{\left(1 + \beta z\sigma^0\right)^2}
\end{aligned}
\tag{A8}
$$

it still remains tedious to derive its expression with respect to both $\sigma^0$ and $z$. This is why we opt for separating the respective effects of type #1 and type #2 efficacies.

First, let us ask what would be the MCD-optimal effort $\hat{z}$ and confidence $\bar{P}_c(\hat{z})$ when $\gamma = 0$, that is, if the only effect of allocating resources is to increase the precision of value representations. We call this the '$\beta$-effect'. In this case, $E[|\Delta\mu||z] = |\Delta\mu^0|$ and $V[|\Delta\mu||z] = 0$ irrespective of $z$. This greatly simplifies *Equations A6–A8*:

$$
\begin{aligned}
\frac{\partial \bar{P}_c(z)}{\partial |\Delta\mu^0|}\Big|_{\gamma=0} &= 6K(z)\sigma(z) \\
\frac{\partial \bar{P}_c(z)}{\partial \sigma^0}\Big|_{\gamma=0} &= -\frac{3K(z)|\Delta\mu^0|}{\left(1 + \beta z\sigma^0\right)^2} \\
K(z)|_{\gamma=0} &= \frac{\pi \bar{P}_c(z)(1 - \bar{P}_c(z))}{(6\sigma(z))^{\frac{3}{2}}}
\end{aligned}
\tag{A9}
$$

Inserting *Equation A9* back into *Equation A5* now yields:

$$
\begin{cases}
\dfrac{\partial \hat{z}}{\partial |\Delta\mu^0|}\Big|_{\gamma=0} = \dfrac{\beta K(\hat{z})\sigma(\hat{z}) - \frac{\partial K(z)}{\partial z}\big|_{z=\hat{z}}}{\frac{\partial K(z)}{\partial |\Delta\mu^0|}\big|_{z=\hat{z}}} \\[2em]
\dfrac{\partial \hat{z}}{\partial \sigma^0}\Big|_{\gamma=0} = \dfrac{2K(\hat{z})\beta\sigma^0 - \frac{\partial K(z)}{\partial z}\big|_{z=\hat{z}}}{2K(\hat{z})\beta\hat{z} - \frac{\partial K(z)}{\partial \sigma^0}\big|_{z=\hat{z}}}
\end{cases}
\tag{A10}
$$

Now the sign of the gradients of $\hat{z}$ w.r.t. $\sigma^0$ and $|\Delta\mu^0|$ are driven by the numerators of **Equation A10** because all partial derivatives of $K(z)$ have unambiguous signs:

$$
\frac{\partial K(z)}{\partial |\Delta\mu^0|}\bigg|_{\gamma=0} = \frac{6\pi(1-2\bar{P}_c(z))K(z)}{(6\sigma(z))^{\frac{1}{2}}} \geq 0
$$

$$
\frac{\partial K(z)}{\partial \sigma^0}\bigg|_{\gamma=0} = -\frac{\pi}{(1+\beta z\sigma^0)^2(6\sigma(z))^{\frac{3}{2}}}\left(6(1-2\bar{P}_c(z))K(z)|\Delta\mu^0| + \frac{\bar{P}_c(z)(1-\bar{P}_c(z))}{4\sigma(z)^2}\right) \leq 0 \qquad \text{(A11)}
$$

$$
\frac{\partial K(z)}{\partial z}\bigg|_{\gamma=0} = \beta K(z)\sigma(z)\left(\frac{1}{4} + \frac{6\pi(1-2\bar{P}_c(z))|\Delta\mu^0|}{(6\sigma(z))^{\frac{3}{2}}}\right) \geq 0
$$

Replacing the expression for $\partial K(z)/\partial z$ in **Equation A11** into **Equation A10** now yields:

$$
\begin{cases}
\dfrac{\partial \hat{z}}{\partial |\Delta\mu^0|}\bigg|_{\gamma=0} \propto 3\beta K(\hat{z})\sigma(\hat{z})\left(\dfrac{1}{4} - \dfrac{2\pi(1-2\bar{P}_c(\hat{z}))|\Delta\mu^0|}{(6\sigma(\hat{z}))^{\frac{3}{2}}}\right) \\[4mm]
\dfrac{\partial \hat{z}}{\partial \sigma^0}\bigg|_{\gamma=0} \propto \beta K(\hat{z})\left(2\sigma^0 - \dfrac{\sigma(\hat{z})}{4} - \dfrac{\pi(1-2\bar{P}_c(\hat{z}))|\Delta\mu^0|}{\sqrt{6\sigma(\hat{z})}}\right)
\end{cases} \qquad \text{(A12)}
$$

At the limit $|\Delta\mu^0| \to 0$, then: $\partial\hat{z}/\partial|\Delta\mu^0| \geq 0$ and $\partial\hat{z}/\partial\sigma^0 \geq 0$. However, one can see from **Equation A12** that there may be a critical value for $|\Delta\mu^0|$, above which the gradient $\partial\hat{z}/\partial|\Delta\mu^0|$ will eventually become negative. This means that the amount of allocated resources will behave as a bell-shaped function of $|\Delta\mu^0|$. This may not be the case along the $\sigma^0$ direction though, because $\sigma^0 \geq \sigma(z)$ and the last term in the brackets shrinks as $\sigma^0$ increases.

Similar derivations eventually yield expressions for the gradients of MCD-optimal confidence:

$$
\begin{cases}
\dfrac{d\bar{P}_c(\hat{z})}{d|\Delta\mu^0|}\bigg|_{\gamma=0} = 3K(\hat{z})\sigma(\hat{z})\left(2 + \beta|\Delta\mu^0|\sigma(\hat{z})\dfrac{\partial\hat{z}}{\partial|\Delta\mu^0|}\right) \\[4mm]
\dfrac{d\bar{P}_c(\hat{z})}{d\sigma^0}\bigg|_{\gamma=0} = 6K(\hat{z})|\Delta\mu^0|\left(\beta\sigma(\hat{z})^2\dfrac{\partial\hat{z}}{\partial\sigma^0} - \dfrac{1}{(1+\beta\hat{z}\sigma^0)^2}\right)
\end{cases} \qquad \text{(A13)}
$$

**Equation A13** implies that, under moderate type #1 efficacy ($\beta \approx 0$), MCD-optimal confidence decreases when $|\Delta\mu^0|$ decreases and/or when $\sigma^0$ increases, irrespective of the amount $\hat{z}$ of allocated resources. In other terms, variations in choice confidence are dominated by variations in the discriminability of prior value representations.

This analysis is exemplified in **Appendix 1—figure 2**, which summarizes the β-effect, in terms of how MCD-optimal resource allocation and choice confidence depend upon $|\Delta\mu^0|$ and $\sigma^0$.

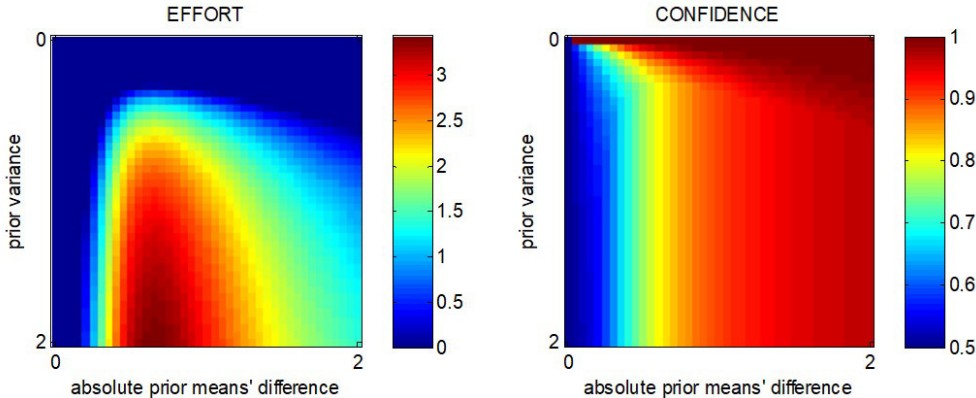

**Appendix 1—figure 2.** The β-effect: MCD-optimal effort and confidence when effort has no impact on the value difference. MCD-optimal effort (left) and confidence (right) are shown as a function of the absolute prior mean difference $|\mu^0|$ (x-axis) and prior variance $\sigma^0$ (y-axis).

One can see that, overall, increasing the prior variance $\sigma^0$ increases the resource demand, which eventually increases the MCD-optimal allocated effort $\hat{z}$. This, however, does not overcompensate for the loss of confidence incurred when increasing the prior variance. This is why the MCD-optimal confidence $\bar{P}_c(\hat{z})$ decreases with the prior variance $\sigma^0$. Note that, for the same reason, the MCD-optimal confidence increases with the absolute prior means' difference $|\Delta\mu^0|$.

Now the impact of the absolute prior means' difference $|\Delta\mu^0|$ on $\hat{z}$ is less trivial. In brief, when $|\Delta\mu^0|$ is high, the MCD-optimal allocated effort $\hat{z}$ decreases when $|\Delta\mu^0|$ increases. This is due to the fact that the resource demand decreases with $|\Delta\mu^0|$. However, there is a critical value for $|\Delta\mu^0|$, below which the MCD-optimal allocated effort $\hat{z}$ *increases* with $|\Delta\mu^0|$. This is because, although the resource demand still increases when $|\Delta\mu^0|$ decreases, the cost of allocating resources overcompensates the gain in confidence. For such difficult decisions, the system does not follow the demand anymore, and progressively de-motivates the allocation of resources as $|\Delta\mu^0|$ continues to decrease. In brief, the amount $\hat{z}$ of allocated resources decreases away from a 'sweet spot', which is the absolute prior means' difference that yields the maximal confidence gain per effort unit. Critically, the position of this sweet spot along the $|\Delta\mu^0|$ dimension decreases with $\beta$ and increases with $\alpha$. This is because confidence gain increases, by definition, with effort efficacy, whereas it becomes more costly when $\alpha$ increases.

Second, let us ask what would be the MCD-optimal effort $\hat{z}$ and confidence $\bar{P}_c(\hat{z})$ when $\beta = 0$, that is, if the only effect of allocating resources is to perturb the value difference. The ensuing '$\gamma$ - effect' is depicted in *Appendix 1—figure 3*.

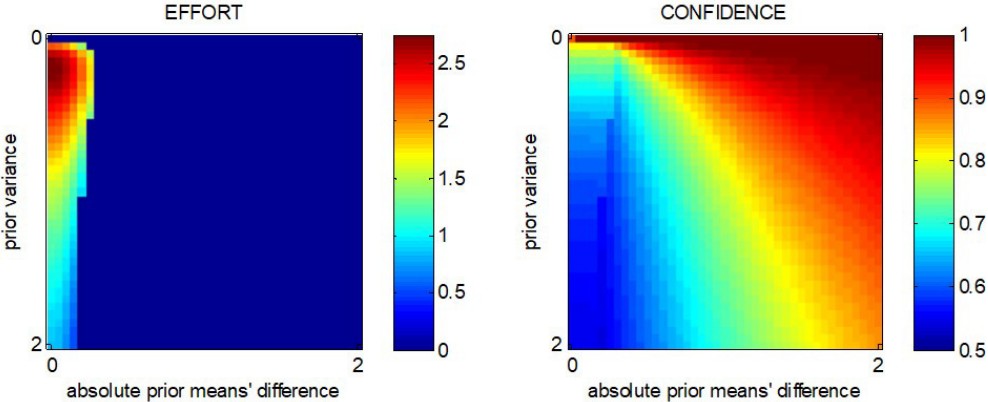

**Appendix 1—figure 3.** The $\gamma$-effect: MCD-optimal effort and confidence when effort has no impact on value precision. Same format as *Appendix 1—figure 2*.

In brief, the overall picture is reversed, with a few minor differences. One can see that increasing the absolute prior means' difference $|\Delta\mu^0|$ decreases the resource demand, which eventually decreases the MCD-optimal allocated effort $\hat{z}$. This can decrease confidence, if $\gamma$ is high enough to overcompensate the effect of variations in $|\Delta\mu^0|$. When no effort is allocated, however, confidence is driven by $|\Delta\mu^0|$, that is, it becomes an increasing function of $|\Delta\mu^0|$. In contrast, variations in the prior variance $\sigma^0$ always overcompensate the ensuing changes in effort, which is why confidence always decreases with $\sigma^0$. In addition, the amount $\hat{z}$ of allocated resources decreases away from a sweet prior variance spot, which is the prior variance $\sigma^0$ that yields the maximal confidence gain per effort unit. Critically, the position of this sweet spot increases with $\gamma$ and decreases with $\alpha$, for reasons similar to the $\beta$-effect.

Now one can ask what happens in the presence of both the $\beta$-effect and the $\gamma$-effect. If the effort unitary cost $\alpha$ is high enough, the MCD-optimal effort allocation is essentially the superposition of both effects. This means that there are two 'sweet spots': one around some value of $|\Delta\mu^0|$ at high $\sigma^0$ ($\beta$-effect) and one around some value of $\sigma^0$ at high $|\Delta\mu^0|$ ($\gamma$-effect). If the effort unitary cost $\alpha$ decreases, then the position of the $\beta$-sweet spot increases and that of the $\beta$-sweet spot decreases, until they effectively merge together. This is exemplified in *Appendix 1—figure 4*.

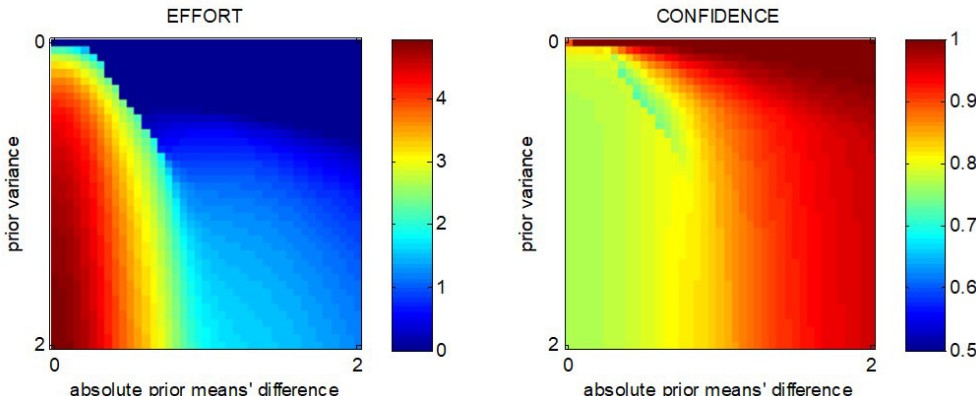

**Appendix 1—figure 4.** MCD-optimal effort and confidence when both types of effort efficacy are operant. Same format as *Appendix 1—figure 2*.

One can see that, somewhat paradoxically, the effort response is now much simpler. In brief, the MCD-optimal effort allocation $\hat{z}$ increases with the prior variance $\sigma^0$ and decreases with the absolute prior means' difference $|\Delta\mu^0|$. The landscape of the ensuing MCD-optimal confidence level $\bar{P}_c(\hat{z})$ is slightly less trivial, but globally, it can be thought of as increasing with $|\Delta\mu^0|$ and decreasing with $\sigma^0$. Here again, this is because variations in $|\Delta\mu^0|$ and/or $\sigma^0$ almost always overcompensate the ensuing effects of changes in allocated effort.

## 3. On MCD parameter estimation

Let $y_t$ be a $6 \times 1$ vector composed of measured choice confidence, spreading of alternatives, value certainty gain, change of mind, response time, and subjective effort rating on trial $t$. Let $u_t$ be a $4 \times 1$ vector, whose two first entries are composed of pre-choice value difference ($\Delta\mathrm{VR}^0$) and average value certainty ($\mathrm{VCR}^0$) ratings, and whose two last entries encode consequential and penalized trials. Finally, let $\varphi$ be the set of unknown MCD parameters (i.e., intrinsic effort cost $\alpha$ and effort efficacies $\beta$ and $\gamma$), augmented with condition-effect parameters and affine transform parameters (see below). From a statistical perspective, the MCD model then reduces to the following observation equation:

$$\bar{y}_t = g(\varphi, u_t) + \varepsilon_t \tag{A14}$$

where $\bar{y}$ denotes data that have been z-scored across trials, $\varepsilon_t$ are model residuals, and the observation mapping $g(\varphi, u_t)$ is given by:

$$g(\varphi, u_t) = \begin{bmatrix} a_1 + b_1 \times s\left( \dfrac{\pi E[|\Delta\mu||\hat{z}]}{\sqrt{3\left(\frac{2}{1/\sigma^0 + \beta\hat{z}} + V[|\Delta\mu||\hat{z}]\right)}} \right) \\ a_2 + b_2 \times \sqrt{\dfrac{\gamma\hat{z}}{\pi}} exp\left( -\dfrac{|\Delta\mu^0|^2}{4\gamma\hat{z}} \right) \\ a_3 + b_3 \times s\left( -\dfrac{\pi|\Delta\mu^0|}{\sqrt{6\gamma\hat{z}}} \right) \\ a_4 + b_4 \times \beta\hat{z} \\ a_5 + b_5 \times \hat{z} \\ a_6 + b_6 \times \hat{z} \end{bmatrix} \tag{A15}$$

where $E[|\Delta\mu||\hat{z}]$ and $V[|\Delta\mu||\hat{z}]$ depend upon $\gamma$ (see *Equations 7 and 8* in the main text). In *Equation A15*, $a_{1:6}$ and $b_{1:6}$ are the unknown offset and slope parameters of the (nuisance) affine transform on MCD outputs. Note that when fitting the MCD model to empirical data, theoretical pre-choice value difference and value certainty ratings are replaced by their empirical proxies, that is, $\Delta\mu^0 \approx \Delta\mathrm{VR}^0$ and $1/\sigma^0 \approx \mathrm{VCR}^0$. In turn, given MCD parameters, *Equations A14 and A15* predict trial-by-trial variations in choice confidence, spreading of alternatives, value certainty gain,

change of mind, response time, and subjective effort rating from variations in prior moments of value representations. We note that *Equation A15* does not yet include condition-specific effects. As we will see, it will be easier to complete the definition of model parameters $\varphi$ once we have explained the variational Laplace scheme for parameter estimation.

Recall that the variational Laplace scheme is an iterative algorithm that indirectly optimizes an approximation to both the model evidence $p(y|m, u)$ and the posterior density $p(\varphi|y, m, u)$, where $m$ is the so-called generative model (i.e., the set of assumptions that are required for inference). The key trick is to decompose the log model evidence into:

$$lnp(y|m, u) = F(q) + D_{KL}(q(\varphi); p(\varphi|y, m, u)) \tag{A16}$$

where $q(\varphi)$ is any arbitrary density over the model parameters, $D_{KL}$ is the Kullback-Leibler divergence and the so-called *free energy* $F(q)$, defined as:

$$F(q) = \langle lnp(\varphi|m) + lnp(y|\varphi, m, u)\rangle_q + S(q) \tag{A17}$$

where $S(q)$ is the Shannon entropy of $q$ and the expectation $\langle\cdot\rangle_q$ is taken under $q$.

From *Equation A16*, maximizing the functional $F(q)$ w.r.t. $q$ indirectly minimizes the Kullback-Leibler divergence between $q(\varphi)$ and the exact posterior $p(\varphi|y, m)$. This decomposition is complete in the sense that if $q(\varphi) = p(\varphi|y, m)$, then $F(q) = lnp(y|m)$.

The variational Laplace algorithm iteratively maximizes the free energy $F(q)$ under simplifying assumptions (see below) about the functional form of $q$, rendering $q$ an approximate posterior density over model parameters and $F(q)$ an approximate log model evidence (*Daunizeau, 2017a*; *Friston et al., 2007*). The free energy optimization is then made with respect to the sufficient statistics of $q$, which makes the algorithm generic, quick, and efficient.

Under normal i.i.d. model residuals (i.e., $\varepsilon_t \sim N(0, 1/\lambda)$), the likelihood function writes:

$$\begin{aligned} p(y|\varphi, \lambda, m, u) &= \prod_t p(y_t|\varphi, \lambda, m, u_t) \\ &= \prod_t N\left(g(\varphi, u_t), \frac{1}{\lambda}I\right) \end{aligned} \tag{A18}$$

where $\lambda$ is the residuals' precision or inverse variance hyperparameter and the observation mapping $g(\varphi, u_t)$ is given in *Equation A15*.

We also use Gaussian priors $p(\varphi|m) = N(\eta_0, \Sigma_0)$ for model parameters and gamma priors for precision hyperparameters $p(\lambda|m) = Ga(\varpi_0, \kappa_0)$.

In what follows, we derive the variational Laplace algorithm under a 'mean-field' separability assumption between parameters and hyperparameters, that is: $q(\varphi, \lambda) = q(\varphi)q(\lambda)$. We will see that this eventually yields a Gaussian posterior density $q(\varphi) \approx N(\eta, \Sigma)$ on model parameters, and a Gamma posterior density $q(\lambda) = Ga(\varpi, \kappa)$ on the precision hyperparameter.

First, let us note that, under the Laplace approximation, the free energy bound on the log-model evidence can be written as:

$$\begin{aligned} F(q) &= \langle I(\varphi)\rangle_{q(\varphi)} + S(q(\varphi)) + S(q(\lambda)) \\ &\approx I(\eta) + \frac{1}{2}ln|\Sigma| + \frac{n_\varphi}{2}ln2\pi + \varpi - ln\kappa + log\Gamma(\varpi) + (1 - \varpi)\psi(\varpi) \end{aligned} \tag{A19}$$

where $n_\varphi$ is the number of parameters, $\Gamma(\cdot)$ is the gamma function, $\psi(\cdot)$ is the digamma function, and $I(\varphi)$ is defined as:

$$I(\varphi) = \langle logp(\varphi|m) + logp(y|\varphi, \lambda, m, u) + logp(\lambda|m)\rangle_{q(\lambda)} \tag{A20}$$

Given the Gamma posterior $q(\lambda)$ on the precision hyperparameter, $I(\varphi)$ can be simply expressed as follows:

$$I(\varphi) = -\frac{1}{2}(\varphi - \eta_0)^T \Sigma_0^{-1}(\varphi - \eta_0) - \frac{\langle\lambda\rangle}{2}\sum_t (y_t - g(\varphi, u_t))^T(y_t - g(\varphi, u_t)) \tag{A21}$$

where we have ignored the terms that do not depend upon $\varphi$, and $\langle\lambda\rangle = E[\lambda|y, m] = \varpi/\kappa$ is the posterior mean of the data precision hyperparameter $\lambda$.

The variational Laplace update rule for the approximate posterior density $q(\varphi)$ on model parameters now simply reduces to an update rule for its sufficient statistics:

$$q(\varphi) \approx N(\eta, \Sigma): \begin{cases} \eta = \arg\max_{\varphi} I(\varphi) \\ \Sigma = -\left[\frac{\partial^2 I}{\partial \varphi^2}|_{\eta}\right]^{-1} \end{cases}$$ (A22)

In *Equation A22*, the first-order moment $\eta$ of $q(\varphi)$ is obtained from the following Gauss-Newton iterative gradient ascent scheme:

$$\eta \leftarrow \eta - \left[\frac{\partial^2 I}{\partial \varphi^2}|_{\eta}\right]^{-1} \frac{\partial I}{\partial \varphi}|_{\eta}$$ (A23)

where the gradient and Hessians of $I(\varphi)$ are given by:

$$\begin{aligned} \frac{\partial I}{\partial \varphi} &= \Sigma_0^{-1}(\eta_0 - \varphi) + \langle\lambda\rangle \frac{\partial g}{\partial \varphi}^T \sum_t (y_t - g(\varphi, u_t)) \\ \frac{\partial^2 I}{\partial \varphi^2} &\approx -\Sigma_0^{-1} - \langle\lambda\rangle \sum_t \frac{\partial g}{\partial \varphi}^T \frac{\partial g}{\partial \varphi} \end{aligned}$$ (A24)

At convergence of the above gradient ascent, the approximate posterior density $q(\varphi)$ on the precision hyperparameter is updated as follows:

$$q(\lambda) = Ga(\varpi, \kappa): \begin{cases} \varpi = \varpi_0 + 3n_t - 1 \\ \kappa = \kappa_0 + \frac{1}{2}\sum_t (y_t - g(\eta, u_t))^T (y_t - g(\eta, u_t)) + tr\left[\frac{\partial g}{\partial \varphi}|_\eta^T \frac{\partial g}{\partial \varphi}|_\eta \Sigma^{-1}\right] \end{cases}$$ (A25)

where $n_t$ is the number of trials.

The variational Laplace scheme alternates between *Equations A22 and A25* iteratively until convergence of the free energy.

Now, let us complete the definition of the model parameter vector $\varphi = \varphi_{1:17}$.

First, note that effort efficiency parameters are necessarily positive. Enforcing this constraint can be done using the following simple change of variable in *Equation A15*: $\beta = exp(\varphi_1)$ and $\gamma = exp(\varphi_2)$. In other words, $\varphi_{1:2}$ effectively measure efficiency parameters in log-space. Second, recall that we want to insert condition-specific effects in the model. More precisely, we expect 'consequential' decisions to be more important than 'neutral' ones, and 'penalized' decisions effectively include an extraneous cost-of-time term. One can model the former condition effect by making $R$ (*Equation 2* in the main text) sensitive to whether the decision is consequential ($u^{(c)} = 1$) or not ($u^{(c)} = 0$), that is: $R_t = exp\left(\varphi_3 u_t^{(c)}\right)$, where $t$ indexes trials, and $\varphi_3$ is the unknown weight of consequential choices on decision importance. This parameterization makes decision importance necessarily positive, and forces non-consequential trials to act as reference choices (in the sense that their decision importance is set to 1). We proxy the latter condition effect by making the effort unitary cost a function of whether the decision is penalized ($u^{(p)} = 1$) or not ($u^{(p)} = 0$), that is: $\alpha_t = exp\left(\varphi_4 + \varphi_5 u_t^{(p)}\right)$, where $\varphi_4$ is the unknown intrinsic effort cost (in log-space), and $\varphi_5$ is the unknown weight of penalized choices on effort cost. The remaining parameters $\varphi_{6:17}$ lump the offsets ($a_{1:6}$) and log-slopes ($logb_{1:6}$: this enforces a positivity constraint on slope parameters) of the affine transform.

Finally, we set the prior probability density functions on model parameters and hyperparameters as follows:

- $p(\varphi_i|m) = N(0, 10^2) \forall i$, that is, the prior mean of model parameters is $\eta_0 = 0$ and their prior variance is $\Sigma_0 = 10^2 \times I$.
- $p(\lambda|m) = Ga(1, 1)$. Since the data has been z-scored prior to model inversion, this ensures that the prior and likelihood components of $I(\varphi)$ are balanced when the variational Laplace algorithm starts.

This completes the description of the variational Laplace approach to MCD inversion. For more details, we refer the interested reader to the existing literature on variational approaches to

approximate Bayesian inference (*Beal, 2003*; *Daunizeau, 2017b*; *Friston et al., 2007*). We note that the above variational Laplace approach is implemented in the opensource VBA toolbox (*Daunizeau et al., 2014*).

In what follows, we use Monte-Carlo numerical simulations to evaluate the ability of this approach to recover MCD parameters. Our parameter recovery analyses proceed as follows. First, we sample a set of model parameters $\varphi$ under a standard i.i.d. normal distribution. Here, we refer to $\varphi_{ij}$ as $i^{th}$ element of $\varphi$ at the $j^{th}$ Monte-Carlo simulation. Second, for each of these parameter set $\varphi_{\cdot j}$, we simulate a series of N=100 decision trials according to *Equation A14 and A15* above (under random prior moments of value representations). Note that we set the variance of model residuals ($\varepsilon$ in *Equation A14*) to match the average correlation between MCD predictions and empirical data (about 20%, see *Figure 4* in the main text). We also used the same rate of neutral, consequential, and penalized choices as in our experiment. Third, we fit the model to the resulting simulated data (after z-scoring) and extract parameter estimates $\eta_{\cdot j}$ (at convergence of the variational Laplace approach). We repeat these three steps 1000 times, yielding a series of 1000 simulated parameter sets, and their corresponding 1000 estimated parameters sets. Should $\eta_{\cdot j} \approx \varphi_{\cdot j} \, \forall j$, then parameter recovery would be perfect. *Appendix 1—figure 5* compares simulated and estimated parameters to each other across Monte-Carlo simulations. Note that we only report recovery results for $\varphi_{1:5}$, since we do not care about nuisance affine transform parameters.

We also quantify pairwise non-identifiability issues, which arise when the estimation method confuses two parameters with each other. We do this using the so-called 'recovery matrices', which summarize whether variations (across the 1000 Monte-Carlo simulations) in estimated parameters faithfully capture variations in simulated parameters. We first z-score simulated and estimated parameters across Monte-Carlo simulations. We then regress each estimated parameter against all simulated parameters through the following multiple linear regression model:

$$\eta_{ij} = \sum_{i'=1}^{5} \theta_{ii'} \varphi_{i'j} + \varsigma_{ij} \tag{A26}$$

where $\theta_{ii'}$ are regression weights, and $\varsigma_{ij}$ are regression residuals. Here, regression weights are partial correlation coefficients between simulated and estimated parameters (across Monte-Carlo simulations). More precisely, $\theta_{ii'}$ quantifies the impact that variations of the simulated parameter $\varphi_{i'\cdot}$ have on variations of the estimated parameter $\eta_{i\cdot}$, conditional on all other simulated parameters. Would parameters be perfectly identifiable, then $\theta_{ii} \approx 1$ and $\theta_{ii'} \approx 0 \, \forall i' \neq i$. Pairwise non-identifiability issues arise when $\theta_{ii'} \neq 0$. In other words, the regression model in *Equation A26* effectively decomposes the observed variability in the series of estimated parameter $\eta_{i\cdot}$ into 'correct variations' that are induced by variations in the corresponding simulated parameter $\varphi_{i\cdot}$, and 'incorrect variations' that are induced by the remaining simulated parameters $\varphi_{i'\cdot}$ (with $i' \neq i$). This analysis is then summarized in terms of 'recovery matrices', which simply report the squared regression weights $\theta_{ii'}^2$ for each simulated parameter (see right panel of *Appendix 1—figure 5*).

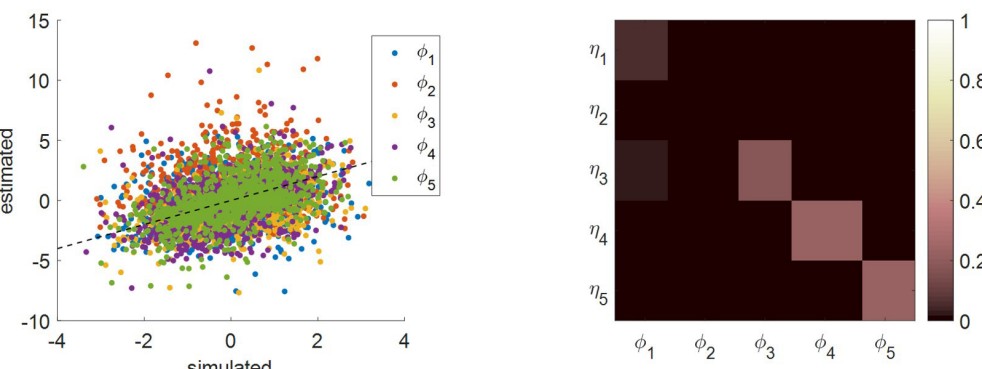

**Appendix 1—figure 5.** Comparison of simulated and estimated MCD parameters. Left panel: estimated parameters (y-axis) are plotted against simulated parameters (x-axis). Each dot is a

*Appendix 1—figure 5 continued on next page*

*Appendix 1—figure 5 continued*

Monte-Carlo simulation and different colors indicate distinct parameters (blue: efficacy type #1, red: efficacy type #2, yellow: unknown weight of consequential choices on decision importance, violet: intrinsic cost of effort, green: unknown weight of penalized choices on effort cost). The black dotted line indicates the identity line (perfect estimation). Right panel: Parameter recovery matrix: each line shows the squared partial correlation coefficient between a given estimated parameter and each simulated parameter (across 1000 Monte-Carlo simulations). Diagonal elements of the recovery matrix measure 'correct estimation variability', i.e. variations in the estimated parameters that are due to variations in the corresponding simulated parameter. In contrast, non-diagonal elements of the recovery matrix measure 'incorrect estimation variability', that is, variations in the estimated parameters that are due to variations in other parameters. Perfect recovery would thus exhibit a diagonal structure, where variations in each estimated parameter are only due to variations in the corresponding simulated parameter. In contrast, strong non-diagonal elements in recovery matrices signal pairwise non-identifiability issues.

One can see that parameter recovery is far from perfect. This is in fact expected, given the high amount of simulation noise. However, no parameter estimate exhibits any noticeable estimation bias, that is, estimation error is non-systematic and directly results from limited data reliability. Recovery matrices provide further quantitative insight regarding the accuracy of parameter estimation.

First, variability in all parameter estimates is mostly driven by variability in the corresponding simulated parameter (amount of 'correct variability': $\varphi_1$: 5.3%, $\varphi_3$: 17.4%, $\varphi_4$: 22.1%, $\varphi_5$: 22.7%, to be compared with 'incorrect variability' – see below), except for type #1 efficacy ($\varphi_2$: 0.3%). The latter estimate is thus comparatively much less efficient than other MCD parameters. This is because $\beta = exp(\varphi_2)$ only has a limited impact on MCD outputs. Second, there are no strong non-identifiability issues (total amount of 'incorrect invariability' is always below 2.7%, even when including nuisance affine transform parameters $\varphi_{6:17}$), except for type #2 effort efficacy. In particular, the latter estimate may be partly confused with intrinsic effort cost (amount of "incorrect variability" driven by $\varphi_1$: 1.6%).

Having said this, the reliability of MCD parameter recovery is globally much weaker than in the ideal case, where data is not polluted with simulation noise (the amount of 'correct variability' in this case is higher than 95% for all parameters – results not shown). This means that acquiring data of higher quality and/or quantity may significantly improve inference on MCD parameters.

We note that the weak identifiability of type #1 effort efficacy ($\beta$) does not imply that some dependent variables will be less well predicted/postdicted than others. Recall that $\beta$ indirectly influences all dependent variables, through its impact on the optimal amount of allocated resources. Therefore, all dependent variables provide information about $\beta$. Importantly, some dependent variables are more useful than others for estimating $\beta$. If empirical measures of these variables become unreliable (e.g., because they are very noisy), then $\beta$ will not be identifiable. However, the reverse is not true. In fact, in our recovery analysis, we found no difference in postdiction accuracy across dependent variables. Now, the question of whether weak $\beta$ identifiability may explain (out-of-sample) prediction errors regarding the impact of MCD input variables (such as $\Delta VR0$) on dependent variables is more subtle. This is because, by construction, MCD parameters control the way MCD input variables eventually influence dependent variables. As one can see from the analytical derivations in section 2 of this Appendix, the impact of input variables on MCD dependent variables (in particular, the optimal amount of allocated resources) depends upon whether $\beta$ dominates effort efficacy (cf. '$\beta$-effect') or not (cf. '$\gamma$-effect'). For example, if $\beta$ dominates, then the relationship between $\Delta VR^0$ and effort is bell-shaped (cf. Figure S6), whereas it is monotonic if $\beta = 0$ (cf. Figure S7). This means that estimation errors on $\beta$ may confuse the predicted relationship between input variables and MCD dependent variables.

## 4. Data descriptive statistics and sanity checks

Recall that we collect value ratings and value certainty ratings both before and after the choice session. We did this for the purpose of validating specific predictions of the MCD model (in particular: choice-induced preference changes: see *Figure 10* in the main text). It turns out this also enables us

to assess the test–retest reliability of both value and value certainty ratings. We found that both ratings were significantly reproducible (value: mean correlation = 0.88, s.e.m. = 0.01, p<0.001, value certainty: mean correlation = 0.37, s.e.m. = 0.04, p<0.001).

We also checked whether choices were consistent with pre-choice ratings. For each participant, we thus performed a logistic regression of choices against the difference in value ratings. We found that the balanced prediction accuracy was beyond chance level (mean accuracy=0.68, s.e.m.=0.01, p<0.001).

## 5. Does choice confidence moderate the relationship between choice and pre-choice value ratings?

Previous studies regarding confidence in value-base choices showed that choice confidence moderates choice prediction accuracy (*De Martino et al., 2013*). We thus split our logistic regression of choices into high- and low-confidence trials, and tested whether higher confidence was consistently associated with increased choice accuracy. A random effect analysis showed that the regression slopes were significantly higher for high- than for low-confidence trials (mean slope difference = 0.14, s.e.m. = 0.03, p<0.001). For the sake of completeness, the impact of choice confidence on the slope of the logistic regression (of choice onto the difference in pre-choice value ratings) is shown in *Appendix 1—figure 6*.

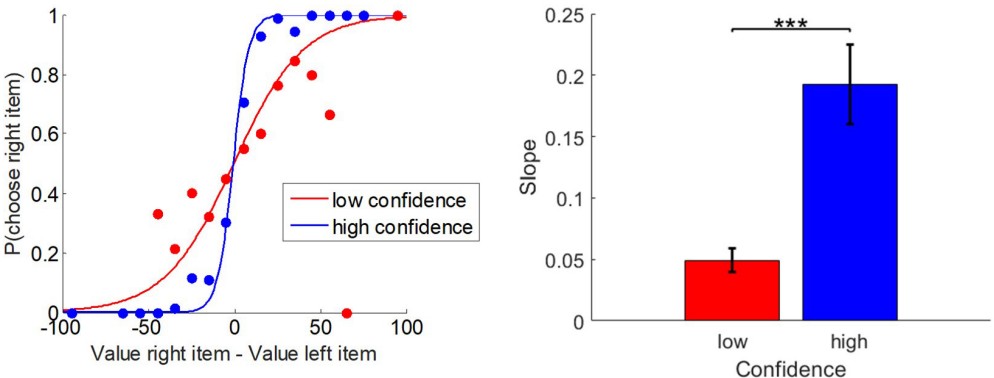

**Appendix 1—figure 6.** Relationship between choices, pre-choice value ratings, and choice confidence. Left panel: the probability of choosing the item on the right (y-axis) is shown as a function of the pre-choice value difference (x-axis), for high- (blue) versus low- (red) confidence trials. The plain lines show the logistic prediction that would follow from group-averages of the corresponding slope estimates. Right panel: the corresponding logistic regression slope (y-axis) is shown for both high- (blue) and low- (red) confidence trials (group means ± s.e.m.).

These results clearly replicate the findings of *De Martino et al., 2013*, which were interpreted with a race model variant of the accumulation-to-bound principle. We note, however, that this effect is also predicted by the MCD model. Here, variations in both (i) the prediction accuracy of choice from pre-choice value ratings and (ii) choice confidence are driven by variations in resource allocation. In brief, the expected magnitude of the perturbation of value representations increases with the amount of allocated resources. This eventually increases the probability of a change of mind. However, although more resources are allocated to the decision, this does not overcompensate for decision difficulty, and thus choice confidence decreases. Thus, low-confidence choices will be those choices that are more likely to be associated with a change of mind. We note that the anti-correlation between choice confidence and change of mind can be seen by comparing *Figures 7* and *8* in the main text.

## 6. How do choice confidence, difference in pre-choice value ratings, and response time relate to each other?

In the main text, we show that trial-by-trial variation in choice confidence is concurrently explained by both pre-choice value and value certainty ratings. Here, we reproduce previous findings relating choice confidence to both absolute value difference $\Delta VR^0$ and response time (*De Martino et al.,*

*2013*). First, for each participant, we regressed response time concurrently against both $|\Delta VR^0|$ and choice confidence. A random effect analysis showed that both have a significant main effect on response time ($\Delta VR^0$: mean GLM beta = $-0.016$, s.e.m. = 0.003, p<0.001; choice confidence: mean GLM beta = $-0.014$, s.e.m. = 0.002; p<0.001), without any two-way interaction (p=0.133). This analysis is summarized in *Appendix 1—figure 7*, together with the full three-way relationship between $|\Delta VR^0|$, confidence, and response time.

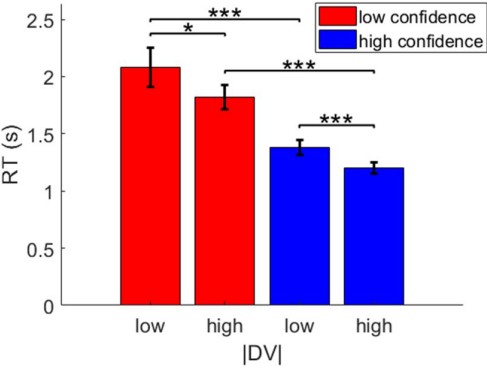 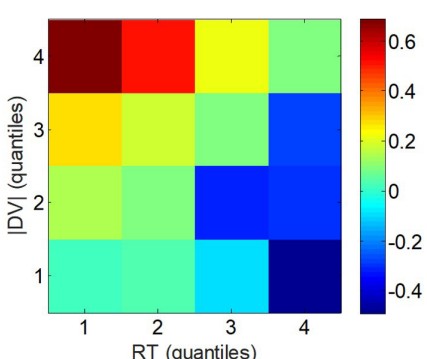

**Appendix 1—figure 7.** Relationship between pre-choice value ratings, choice confidence, and response times. Left panel: response times (y-axis) are plotted as a function of low- and high- $|\Delta VR^0|$ (x-axis) for both low- (red) and high- (blue) confidence trials. Error bars represent s.e.m. Right panel: A heatmap of mean z-scored confidence is shown as a function of both response time (x-axis) and $|\Delta VR^0|$ (y-axis).

In brief, confidence increases with the absolute value difference and decreases with response time. This effect is also predicted by the MCD model, for reasons identical to the explanation of the relationship between confidence and choice accuracy (see above). Recall that, overall, an increase in choice difficulty is expected to yield an increase in response time and a decrease in choice confidence. This would produce the same data pattern as *Appendix 1—figure 7*, although the causal relationships implicit in this data representation is partially incongruent with the computational mechanisms underlying MCD.

## 7. Do post-choice ratings better predict choice and choice confidence than pre-choice ratings?

The MCD model assumes that value representations are modified during the decision process, until the MCD-optimal amount of resources is met. This eventually triggers the decision, whose properties (i.e., which alternative option is eventually preferred, and with which confidence level) then reflect the modified value representations. If post-choice ratings are reports of modified value representations at the time when the choice is triggered, then choice and its associated confidence level should be better predicted with post-choice ratings than with pre-choice ratings. In what follows, we test this prediction.

In Section 4 of this Appendix, we report the result of a logistic regression of choice against pre-choice value ratings (see also *Appendix 1—figure 6*). We performed the same regression analysis, but this time against post-choice value ratings. For each subject, we then measured the ensuing predictive power (here, in terms of balanced accuracy or BA) for both pre-choice and post-choice ratings. The main text also features the result of a multiple linear regression of choice confidence ratings onto $|\Delta VR^0|$ and $VCR^0$ (*Figure 8* in the main text). Again, we performed the same regression, this time against post-choice ratings. For each subject, we then measured the ensuing predictive power (here, in terms of percentage of explained variance or $R^2$) for both pre-choice and post-choice ratings.

A simple random effect analysis shows that the predictive power of post-choice ratings is significantly higher than that of pre-choice ratings, both for choice (mean difference in BA=7%, s.e.m. =0.01, p<0.001) and choice confidence (mean difference in $R^2$=3%, s.e.m.=0.01, p=0.004).

## 8. Analysis of eye-tracking data

We first checked whether pupil dilation positively correlates with participants' subjective effort ratings. We epoched the pupil size data into trial-by-trial time series, and temporally co-registered the epochs either at stimulus onset (starting 1.5 s before the stimulus onset and lasting 5 s) or at choice response (starting 3.5 s before the choice response and lasting 5 s). Data was baseline-corrected at stimulus onset. For each participant, we then regressed, at each time point during the decision, pupil size onto effort ratings (across trials). Time series of regression coefficients were then reported at the group level, and tested for statistical significance (correction for multiple comparison was performed using random field theory 1D-RFT). *Appendix 1—figure 8* summarizes this analysis, in terms of the baseline-corrected time series of regression coefficients.

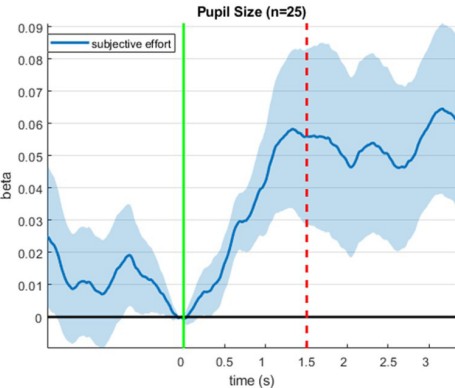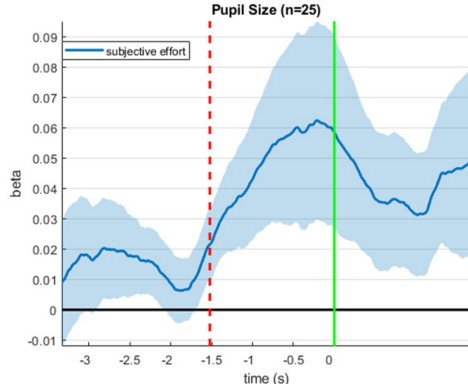

**Appendix 1—figure 8.** Correlation between pupil size and subjective effort ratings during decision time. Left panel: Mean (± s.e.m.) correlation between pupil size and subjective effort (y-axis) is plotted as a function of peristimulus time (x-axis). Here, epochs are co-registered w.r.t. stimulus onset (the green line indicates stimulus onset and the red dotted line indicates the average choice response). Right panel: Same, but for epochs co-registered w.r.t. choice response (the green line indicates choice response and the red dotted line indicates the average stimulus onset).

We found that the correlation between subjective effort ratings and pupil dilation became significant from 500 ms after stimulus onset onwards. Note that, using the same approach, we found a negative correlation between pupil dilation and pre-choice absolute value difference $|\Delta VR^0|$. However, this relationship disappeared when we entered both $|\Delta VR^0|$ and effort into the same regression model.

Our eye-tracking data also allowed us to ascertain which item was being gazed at for each point in peristimulus time (during decisions). Using the choice responses, we classified each time point as a gaze at the (to be) chosen item or at the (to be) rejected item. We then derived, for each decision, the ratio of time spent gazing at chosen/rejected items versus the total duration of the decision (between stimulus onset and choice response). The difference between these two gaze ratios measures the overt attentional bias toward the chosen item. We refer to this as the gaze bias. Consistent with previous studies, we found that chosen items were gazed at more than rejected items (mean gaze bias = 0.02, s.e.m. = 0.01, p=0.067). However, we also found that this effect was in fact limited to low effort choices. *Appendix 1—figure 9* shows the gaze bias for low- and high-effort trials, based on a median-split of subjective effort.

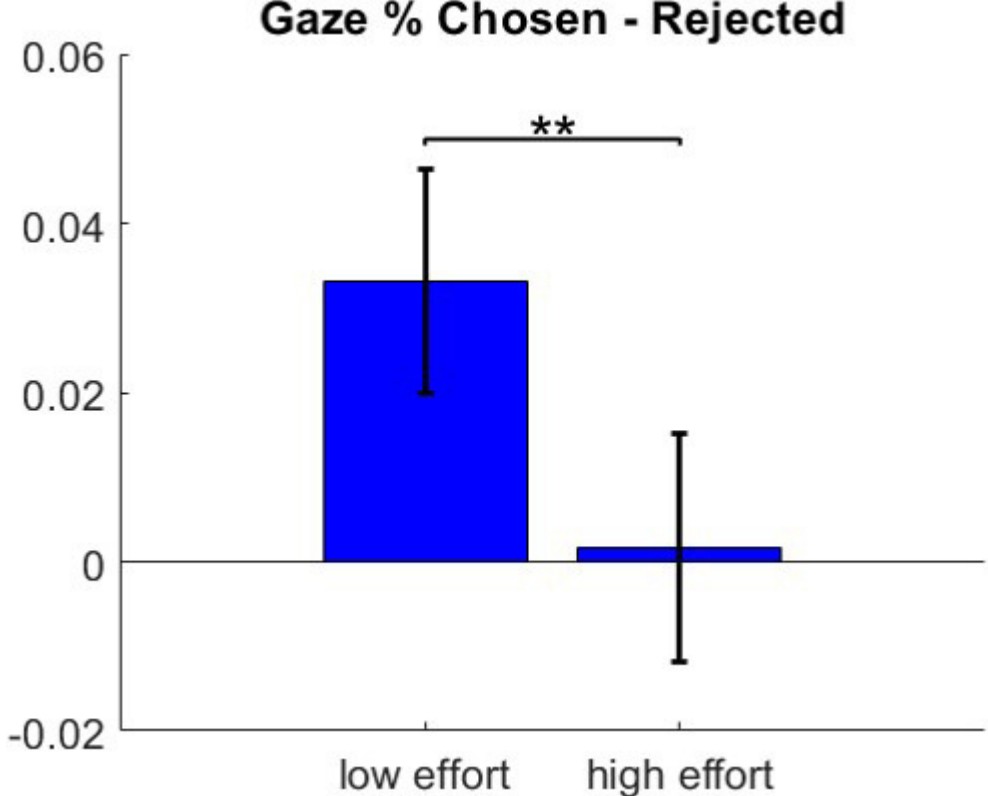

**Appendix 1—figure 9.** Gaze bias for low- and high-effort trials. Mean (± s.e.m.) gaze bias is plotted for both low- (left) and high- (right) effort trials.

We found that there was a significant gaze bias for low effort choices (mean gaze ratio difference = 0.033, s.e.m. = 0.013, p=0.009), but not for high effort choices (mean gaze ratio difference = 0.002, s.e.m. = 0.014, p=0.453). A potential trivial explanation for the fact that the gaze bias is large for low effort trials is that these are the trials where participants immediately recognize their favorite option, which attracts their attention. More interesting is the fact that the gaze bias is null for high effort trials. This may be taken as evidence for the fact that, on average, people allocate the same amount of (attentional) resources to both options. This is important, because we use this simplifying assumption in our MCD model derivations.

## 9. Comparison with evidence-accumulation (DDM) models

In the main text, we evaluate the accuracy of the MCD model predictions, without considering alternative computational scenarios. Here, we report results of a model-based data analysis that relies on the standard drift-diffusion decision or DDM model for value-based decision-making (*De Martino et al., 2013*; *Lopez-Persem et al., 2016*; *Milosavljevic et al., 2010*; *Ratcliff et al., 2016*; *Tajima et al., 2016*).

In brief, DDMs tie together decision outcomes and response times by assuming that decisions are triggered once the accumulated evidence in favor of a particular option has reached a predefined threshold or bound (*Ratcliff and McKoon, 2008*; *Ratcliff et al., 2016*). Importantly here, evidence accumulation has two components: a drift term that quantifies the strength of evidence and a random diffusion term that captures some form of neural perturbation of evidence accumulation. The latter term allows choice outcomes to deviate from otherwise deterministic, evidence-driven, decisions.

Importantly, standard DDMs do not predict choice confidence, spreading of alternatives, value certainty gain, or subjective effort ratings. This is because these concepts have no straightforward definition under the standard DDM. However, DDMs can be used to make out-of-sample trial-by-trial predictions of, for example, decision outcomes, from parameter estimates obtained with response times alone. This enables a straightforward comparison of MCD and DDM frameworks, in

terms of the accuracy of RT 'postdictions' and change of mind out-of-sample prediction. Here, we also make sure both models rely on the same inputs: namely, pre-choice value ($\Delta VR^0$) and value certainty ($VCR^0$) ratings as well as information about task conditions.

The simplest DDM variant includes the following set of five unknown parameters: the drift rate $v$, the bound's height $b$, the standard deviation of the diffusion term $\sigma$, the initial decision bias $x_0$, and the non-decision time $T_{nd}$. Given these model parameters, the expected response time (conditional on the decision outcome) is given by *Srivastava et al., 2016*:

$$E[RT|o,v,x_0,b,\sigma,T_{nd}] = \frac{b}{v}\left(2coth\left(\frac{2vb}{\sigma^2}\right) - \left(1+o\frac{x_0}{b}\right)coth\left(\left(1+o\frac{x_0}{b}\right)\frac{vb}{\sigma^2}\right)\right) + T_{nd} \qquad (A27)$$

where $o \in \{-1,1\}$ is the decision outcome. One can then evaluate Equation A27 at each trial, given its corresponding set of DDM parameters. In particular, if one knows how, for example, drift rates vary over trials, then one can predict the ensuing expected RT variations. In typical applications to value-based decision-making, drift rates are set proportional to the difference $\Delta VR^0$ in value ratings (*De Martino et al., 2013*; *Krajbich et al., 2010*; *Lopez-Persem et al., 2016*; *Milosavljevic et al., 2010*). One can then define a likelihood function for observed response times from the following observation equation: $RT = E[RT|o,v,x_0,b,\sigma,T_{nd}] + \varepsilon$, where $\varepsilon$ are trial-by-trial DDM residuals. The variational Laplace treatment of the ensuing generative model then yields estimates of the remaining DDM parameters.

Out-of-sample predictions of change of mind (i.e., decision errors) can then be derived from DDM parameter estimates (*Bogacz et al., 2006*):

$$\begin{aligned} Q_{DDM} &= P(sign(o) \neq sign(v)|v,b,\sigma,x_0) \\ &= \frac{1}{1+exp\left(\frac{2vb}{\sigma^2}\right)} - \frac{1-exp\left(\frac{2|vx_0|}{\sigma^2}\right)}{exp\left(\frac{2vb}{\sigma^2}\right) - exp\left(\frac{-2vb}{\sigma^2}\right)} \end{aligned} \qquad (A28)$$

where $Q_{DDM}$ is the DDM equivalent to the probability $Q(\hat{z})$ of a change of mind under the MCD model (see *Equation 14* in the main text).

Here, we use two modified variants of the standard DDM for value-based decisions. In all of these variants, we allow the DDM system to change its speed-accuracy tradeoff according to whether the decision is consequential ($u^{(c)} = 1$) or not ($u^{(c)} = 0$), and/or 'penalized' ($u^{(p)} = 1$) or not ($u^{(p)} = 0$). This is done by enabling the decision bound to vary over trials, i.e., $b_t \equiv exp\left(b^{(0)} + b^{(c)}u_t^{(c)} + b^{(p)}u_t^{(p)}\right)$, where $t$ indexes trials. Here, $b^{(0)}$, $b^{(c)}$, and $b^{(p)}$ are unknown parameters that quantify the bound's height of 'neutral' decisions, and the strength of 'consequential' and 'penalized' condition effects, respectively. The exponential mapping is used for imposing a positivity constraint on the resulting bound (see section 8 above). One might then expect that $b^{(c)} > 0$ and $b^{(p)} < 0$, that is, 'consequential' decisions demand more evidence than 'neutral' ones, whereas 'penalized' decisions favor speed over accuracy.

The two DDM variants then differ in terms of how pre-choice value certainty is taken into account (*Lee and Usher, 2020*):

- DDM1: at each trial, the drift rate is set to the affine-transformed certainty-weighted value difference, that is, $v_t \equiv v^{(0)} + v^{(s)} \times VCR_t^0 \times \Delta VR_t^0$, where $v^{(0)}$ and $v^{(s)}$ are unknown parameters that control the offset and slope of the affine transform, respectively. Here, the strength of evidence in favor of a given alternative option is measured in terms of a signal-to-noise ratio on value. Note that the diffusion standard deviation $\sigma$ is kept fixed across trials.
- DDM2: at each trial, the drift rate is set to the affine-transformed value difference, that is, $v_t \equiv v^{(0)} + v^{(s)} \times \Delta VR_t^0$, and the diffusion standard deviation is allowed to vary over trials with value certainty ratings: $\sigma_t \equiv exp\left(\sigma^{(0)} - exp(\sigma^{(1)}) \times VCR_t^0\right)$. Here, $\sigma^{(0)}$ and $\sigma^{(1)}$ are unknown parameters that quantify the fixed and varying components of the diffusion standard deviation, respectively. In this parameterization, value representations that are more certain will be signaled more reliably. Note that the statistical complexity of DDM2 is higher than that of DDM1 (one additional unknown parameter).

For each subject and each DDM variant, we estimate unknown parameters from RT data alone using *Equation A27*, and derive out-of-sample predictions for changes of mind using *Equation A28*.

We then measure the accuracy of trial-by-trial RT postdictions and out-of-sample change of mind predictions, in terms of the correlation between observed and predicted/postdicted variables. We also perform the exact same analysis under the MCD model (this is slightly different from the analysis reported in the main text, because only RT data is included in model fitting here).

To begin with, we compare the accuracy of RT postdictions, which is summarized in *Appendix 1—figure 10*.

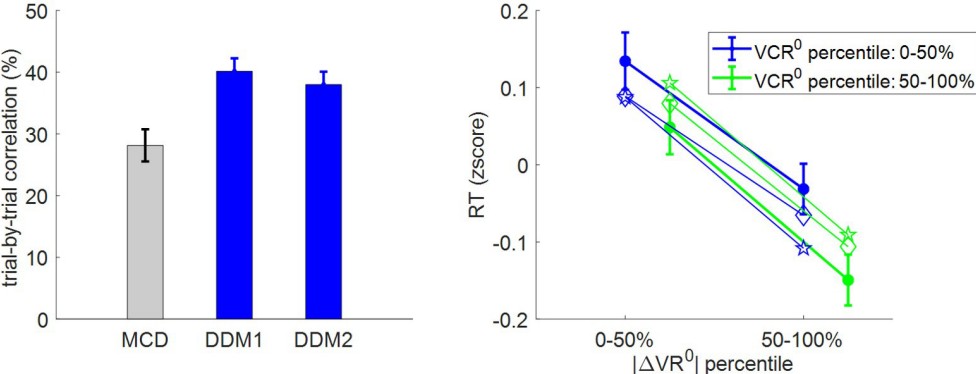

**Appendix 1—figure 10.** Accuracy of RT postdictions. Left panel: The mean within-subject (across-trial) correlation between observed and postdicted RT data (y-axis) is plotted for each model (gray: MCD, blue: DDM1 and DDM2); error bars depict s.e.m. Right panel: Mean z-scored log-RT (y-axis) is shown as a function of $|\Delta VR^0|$ (x-axis) and $VCR^0$ (color code: blue = 0–50% lower quantile, green = 50–100% upper quantile); solid lines indicate empirical data (error bars represent s.e.m.), diamond-dashed lines represent DDM1 postdictions and star-dotted lines show DDM2 postdictions.

One can see that the RT postdiction accuracy of both DDMs is higher than that of the MCD model. In fact, one-sample paired t-tests on the difference between DDM and MCD within-subject accuracy scores show that this comparison is statistically significant (DDM1: mean accuracy difference = 12.3%, s.e.m. = 2.6%, $p < 10^{-3}$; DDM2: mean accuracy difference = 10.5%, s.e.m. = 2.6%, $p < 10^{-3}$; two-sided t-tests). In addition, one can see that DDM1 accurately captures variations in RT data that are induced by $\Delta VR^0$ and $VCR^0$. However, DDM2 is unable to reproduce the impact of $VCR^0$ (cf. wrong effect direction). This is because, in DDM2, as value certainty ratings increase and the diffusion standard deviation decreases, the probability that DDM bounds are hit sooner decreases (hence prolonging RT on average). These results reproduce recent investigations of the impact of value certainty ratings on DDM predictions (*Lee and Usher, 2020*).

Now, *Appendix 1—figure 11* summarizes the accuracy of out-of-sample change of mind predictions.

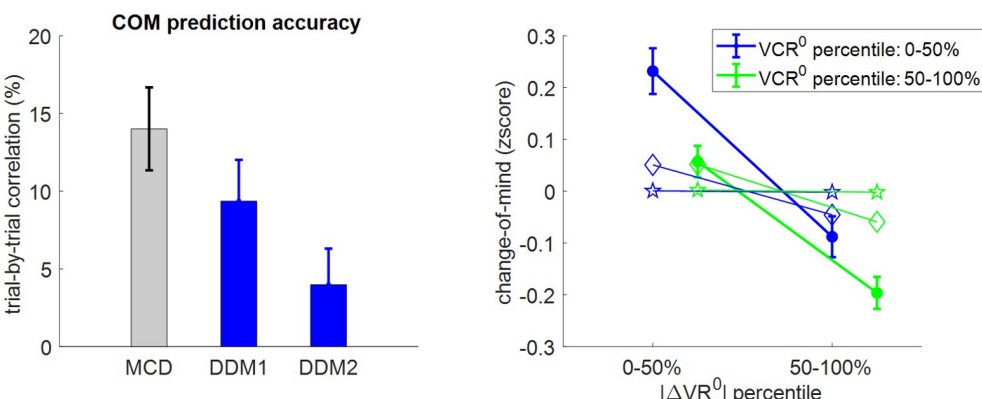

**Appendix 1—figure 11.** Accuracy of out-of-sample change of mind postdictions. Same format as *Appendix 1—figure 10*.

It turns out that the MCD model exhibits the highest accuracy of out-of-sample change of mind predictions. One-sample paired t-tests on the difference between DDM and MCD within-subject

accuracy scores show that this comparison reaches statistical significance for both DDM1 (mean accuracy difference=-5%, s.e.m. = 2.4%, p=0.046; two-sided t-test) and DDM2 (mean accuracy difference = $-9.9\%$, s.e.m. = 3.4%, p=0.006; two-sided t-test). One can also see that neither DDM variant accurately predicts the effects of $\Delta VR^0$ and $VCR^0$.

In brief, the DDM framework might be better than the MCD model at capturing trial-by-trial variations in RT data. This may not be surprising, given the longstanding success of the DDM on this issue (*Ratcliff et al., 2016*). The result of this comparison, however, depends upon how the DDM is parameterized (cf. wrong effect direction of $VCR^0$ for DDM2). More importantly, in our context, DDMs make poor out-of-sample predictions on decision outcomes, at least when compared to the MCD model. For the purpose of predicting decision-related variables from effort-related variables, one would thus favor the MCD framework.

## 10. Accounting for saturating γ-effect

When deriving the MCD model, we considered a linear γ-effect, that is, we assumed that the variance of the perturbation $\delta(z)$ of value representation modes increases linearly with the amount $z$ of allocated resources (*Equation 6* in the main text). However, one might argue that the marginal impact of effort on the variance of $\delta(z)$ may decrease as further resources are allocated to the decision. In other terms, the magnitude of the perturbation (per unit of resources) that one might expect when no resources have yet been allocated may be much higher than when most resources have already been allocated. In turn, *Equation 6* would be replaced by:

$$\begin{aligned} \mu_i(z) &= \mu_i^0 + \delta_i \\ \delta_i &\sim N(0, f(z,\gamma)) \end{aligned} \tag{A29}$$

where the variance $f(z,\gamma)$ of the modes' perturbations would be a saturating function of $z$, e.g:

$$f(z,\gamma) = \gamma_1(1 - exp(-\gamma_2 z)) \tag{A30}$$

where $\gamma_1$ is the maximum or plateau variance that perturbations can exhibit and $\gamma_2$ is the decay rate toward the plateau variance.

It turns out that this does not change the mathematical derivations of the MCD model, that is, model predictions still follow *Equations 9–14* in the main text, having replaced $\gamma z$ with $f(z,\gamma)$ everywhere.

Model simulations with this modified MCD model show no qualitative difference from its simpler variant (linear γ-effect), across a wide range of $\gamma_{1,2}$ parameters. Having said this, the modified MCD model is in principle more flexible than its simpler variant, and may thus exhibit additional explanatory power. We thus performed a formal statistical model comparison to evaluate the potential advantage of considering saturating γ-effects. In brief, we performed the same within-subject analysis as with the simpler MCD variant (see main text). We then measured the accuracy of model post-dictions on each dependent variable and performed a random-effect group-level Bayesian model comparison (*Rigoux et al., 2014*; *Stephan et al., 2009*). The results of this comparison are summarized in *Appendix 1—figure 12*.

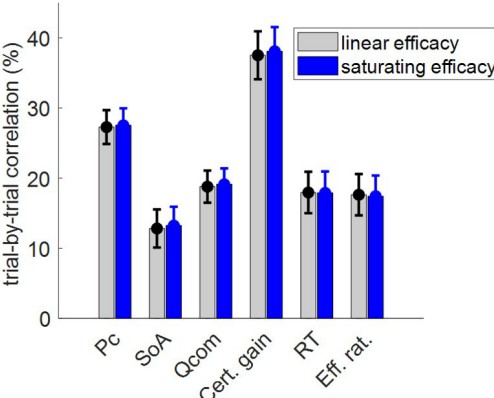
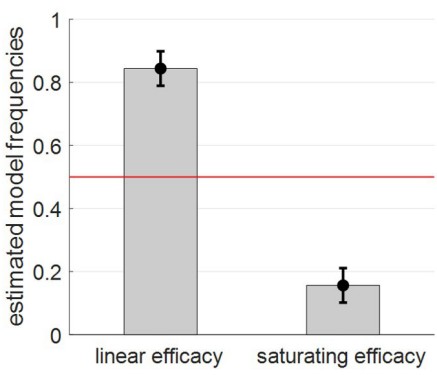

**Appendix 1—figure 12.** Comparisons of MCD model with linear and saturating γ-effects. Left panel: The mean within-subject (across-trial) correlation between observed and postdicted data (y-axis) is plotted for dependent variable (x-axis, from left to right: choice confidence, spreading of alternatives, change of mind, certainty gain, RT and subjective effort ratings) and each model (gray: MCD with linear efficacy, blue: MCD with saturating efficacy); error bars depict s.e.m. Right panel: Estimated model frequencies from the random-effect group-level Bayesian model comparison; error bars depict posterior standard deviations.

First, one can see that considering saturating γ-effects does not provide any meaningful advantage in terms of MCD postdiction accuracy. Second, Bayesian model section clearly favors the simpler (linear γ-effect) MCD variant (linear efficacy: estimated model frequency = 84.4 ± 5.5%, exceedance probability = 1, protected exceedance probability = 0.89). We note that other variants of the MCD model may be proposed, with similar modifications (e.g., nonlinear effort costs, non-Gaussian – skewed – value representations). Preliminary simulations seem to confirm that such modifications would not change the qualitative nature of MCD predictions. In other terms, the MCD model may be quite robust to these kinds of assumptions. Note that these modifications would necessarily increase the statistical complexity of the model (by inserting additional unknown parameters). Therefore, the limited reliability of behavioral data (such as we report here) may not afford subtle deviations to the simple MCD model variant we evaluate here.

## 11. Comparing MCD and model-free postdiction accuracy

The MCD model provides quantitative predictions for both effort-related and decision-related variables, from estimates of three native parameters (effort unitary cost and two types of effort efficacy), which control all dependent variables. However, the model prediction accuracy is not perfect, and one may wonder what is the added value of MCD compared to model-free analyses.

To begin with, recall that one cannot make out-of-sample predictions in a model-free manner (e.g., there is nothing one can learn about effort-related variables from regressions of decision-related variables on $\Delta VR^0$ and $VCR^0$). In contrast, a remarkable feature of model-based analyses is that training the model on some subset of variables is enough to make out-of-sample predictions on other (yet unseen) variables. In this context, MCD-based analyses show that variations in response times, subjective effort ratings, changes of mind, spreading of alternatives, choice confidence, and precision gain can be predicted from each other under a small subset of modeling assumptions.

Having said this, model-free analyses can be used to provide a reference for the accuracy of MCD postdictions. For example, one may regress each dependent variable onto $\Delta VR^0$, $VCR^0$, and indicator variables of experimental conditions (whether or not the choice is 'consequential' and/or 'penalized'), and measure the correlation between observed and postdicted variables. This provides a benchmark against which MCD postdiction accuracy can be evaluated. To enable a fair statistical comparison, we re-performed MCD model fits, this time fitting each dependent variable one by one (leaving the others out). In what follows, we refer to this as 'MCD 1-variable fits'. The results of this analysis are summarized in *Appendix 1—figure 13*:

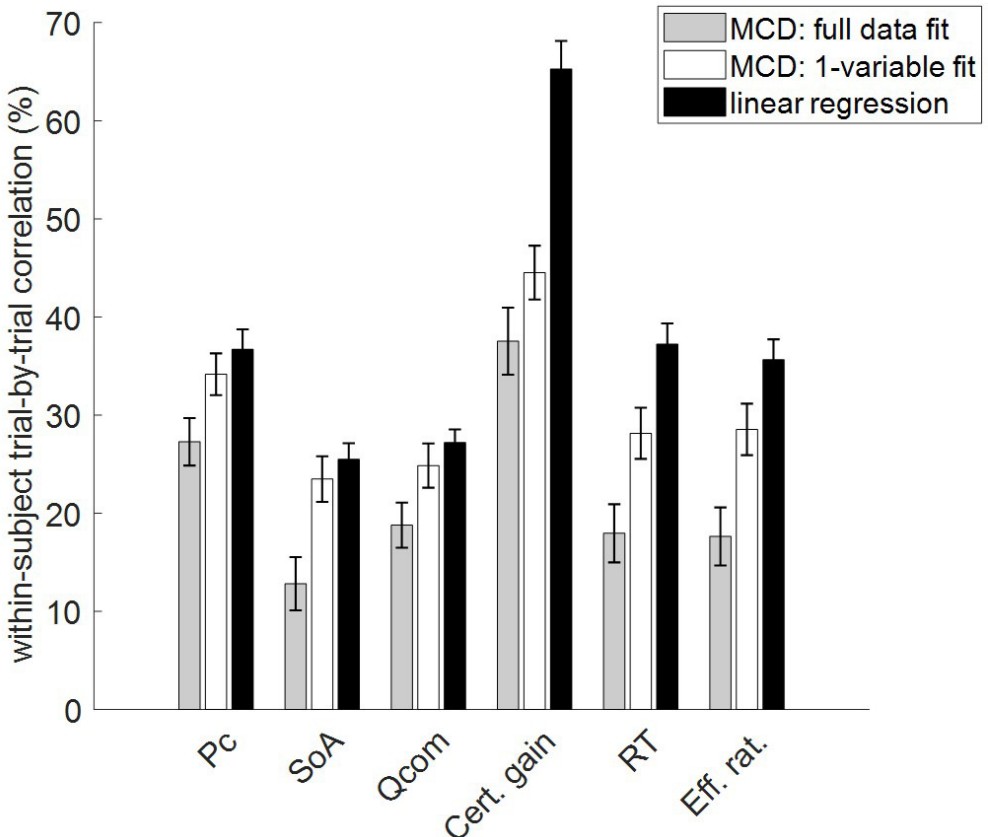

**Appendix 1—figure 13.** Comparisons of MCD and model-free postdiction accuracies. The mean within-subject (across-trial) correlation between observed and postdicted data (y-axis) is plotted for each variable (x-axis, from left to right: choice confidence, spreading of alternatives, change of mind, certainty gain, RT, and subjective effort ratings), and each fitting procedure (gray: MCD full data fit, white: MCD 1-variable fit, and black: linear regression). Error bars depict standard error of the mean.

As expected, MCD 1-variable fits have better postdiction accuracy than the MCD 'full-data' fit. This is because the latter approach attempts to explain all dependent variables with the same parameter set, which requires finding a compromise between all dependent variables.

Now, model-free regressions seem to show globally better postdiction accuracy than MCD 1-variable fits: on average, the MCD model captures about 81% of the variance explained using linear regressions. However, the postdiction accuracy difference is only significant for effort-related variables (RT: p=0.0002, subjective effort rating: p=0.0007), but not for decision-related variables (choice confidence: p=0.06, spreading of alternatives: p=0.28, change of mind: p=0.24) except certainty gain ($p<10^{-4}$).

A likely explanation here is that the MCD model includes constraints that prevent one-variable fits from matching the model-free postdiction accuracy level. In turn, one may want to extend the MCD model with the aim of relaxing these constraints. Having said this, these constraints necessarily derive from the modeling assumptions that enable the MCD model to make out-of-sample predictions. We comment on this and related issues in the Discussion section of the main text.

