## [Decision Letter]

**Acceptance summary:**

This work addresses a timely and heavily debated subject: the role of mental effort in value-based decision-making. Plenty of models attempt to explain value-based choice behavior, and there is a growing number of computational accounts concerning the allocation of mental effort therein. Yet, little theoretical work has been done to relate the two literatures. The current paper contributes a novel and inspiring step in this direction.

**Decision letter after peer review:**

Thank you for submitting your article "Trading Mental Effort for Confidence in the Metacognitive Control of Value-Based Decision-Making" for consideration by *eLife*. Your article has been reviewed by 3 peer reviewers, and the evaluation has been overseen by Tobias Donner as the Reviewing Editor and Michael Frank as the Senior Editor. The following individual involved in review of your submission has agreed to reveal their identity: Andrew Westbrook (Reviewer #3).

The reviewers have discussed the reviews with one another and the Reviewing Editor has drafted this decision to help you prepare a revised submission.

As the editors have judged that your manuscript is of interest, but as described below that additional experiments are required before it is published, we would like to draw your attention to changes in our revision policy that we have made in response to COVID-19 (https://elifesciences.org/articles/57162). First, because many researchers have temporarily lost access to the labs, we will give authors as much time as they need to submit revised manuscripts. We are also offering, if you choose, to post the manuscript to bioRxiv (if it is not already there) along with this decision letter and a formal designation that the manuscript is "in revision at *eLife*". Please let us know if you would like to pursue this option.

Summary:

This manuscript addresses a timely subject: the role of cognitive control (or mental effort) in value-based decision making. While there are plenty of models explaining value-based choice, and there is a growing number of computational accounts concerning effort-allocation, little theoretical work has been done to relate the two literatures. This manuscript contributes a novel and interesting step in this direction, by introducing a computational account of meta-control in value-based decision making. According to this account, meta-control can be described as a cost-benefit analysis that weighs the benefits of allocating mental effort against associated costs. The benefits of mental effort pertain to the integration of value-relevant information to form posterior beliefs about option values. Given a small set of parameters, as well as pre-choice value ratings and pre-choice uncertainty ratings as inputs to the model, it can predict relevant decision variables as outputs, such as choice accuracy, choice confidence, choice induced preference changes, response time and subjective effort ratings. The study fits the model to data from a behavioral experiment involving value-based decisions between food items. The resulting behavioral fits reproduce a number of predictions derived from the model. Finally, the article describes how the model relates to established accumulator models of decision-making.

The (relatively simple) model is impressive in its apparent ability to reproduce qualitative patterns across diverse data including choices, RTs, choice confidence ratings, subjective effort, and choice-induced changes in relative preferences successfully. The model also appears well-motivated, well-reasoned, and well-formulated. While all reviewers agreed that the manuscript is of potential interest, they also all felt that a stronger case needs to be made for the explanatory power of the model, and that the model should be embedded more thoroughly in the existing literature on this topic.

Essential revisions:

1. Evaluation of the (explanatory power of) the model.

1a. Parameter recoverability: Please include an analysis of parameter recoverability: How well can the fitting procedure recover model parameters from data generated by the model?

1b. Fitting procedure: Rather than fitting the model to all dependent variables at once, it would be more compelling to fit the model to a subset of established decision-related variables (e.g. accuracy, choice confidence, choice induced preference changes) and then evaluate if, and how well, the fitted model can predict out-of-sample variables related to effort allocation (e.g. response time and subjective effort ratings). The latter would be a more stringent test of the model, and may serve to highlight its value for linking variables related to value-based decision making to variables related to meta-control.

1c. Model complexity: Assess (through model comparison) how many degrees of freedom are needed to account for the data (e.g. by fixing some of the crucial parameters and evaluating the fit). Currently, the authors show that their model explains more variance in dependent variables when fit to real data than random data. Almost any model which systematically relates independent variables to dependent variables would explain more variance when fit to real data than to data. It would be more useful to know whether (and if so, how much) the model explains data better, than, e.g. a model with where effort only affects precision (β efficacy), or a model in which effort only impacts value mode (γ efficacy).

1d. Single-subject data.

The model appears to do fairly well in predicting aggregate, group-level data, but does it predict subject-level data? Or, does it sometimes make unrealistic predictions when fitting to individual subjects? The Authors should provide evidence of whether it can or cannot describe subject level choices, confidence ratings, subjective effort, etc.

2. Qualify central assumptions underlying the model.

2a. The model assumes that it is "rewarding" to choose the correct (highest-value) option (B = R*P). Is this realistic? If the two options have approx the same value, then R should be small (it doesn't matter which one you choose); if the options have different value, it is important to choose the correct one. Of course, the probability P_c_ continuously differentiates between the two options, but that is not the same as the reward. Can the predictions generalise toward a more general R that depends on value difference?

2b. Is it reasonable to assume that variance would increase as a linear function of resource allocation? It seems to me that variance might increase initially, but then each increment of resources would add diminishing variance to the mode since, e.g., new mnesic evidence should tend to follow old evidence. How sensitive are model predictions to this assumption? What about if each increment of resources added to variance in an exponentially decreasing fashion? What about anchoring biases? Because anchoring biases suggest that we estimate things with reference to other value cues, should we always expect that additional resources increase the expected value difference, or might additional effort actually yield smaller value differences over time? If we relax this assumption, how does this impact model predictions?

3. Address relationship to other accounts.

3a. Does the current model predict the diverse dependent variables better than a standard accumulator models of decision-making?

3b. The model could also situate itself better in the broader existing literature on the topic. For instance, how does the model compares to existing computational work on this matter, e.g. the models described in Izuma and Murayama (2013) or the efficient coding account of Polanía, Woodford, and Ruff (2019)? We understand that the presented model can account for some phenomena that the other models cannot account for, at least without auxiliary assumptions (e.g. subjective effort ratings), but the interested reader might want to know how well the presented model can explain established decision-related variables, such as decision confidence, choice accuracy or choice-induced preference changes compared to existing models, by having them contrasted in a formal manner. Finally, it would seem fair to relate the presented account to emerging, more mechanistically explicit accounts of meta-control in value-based decision making (e.g. Callaway, Rangel and Griffiths, 2020; Jang, Sharma, and Drugowitsch, 2020). Ideally, some of the above would be addressed in the form of formal model comparisons, but we realise that this may be difficult to achieve in practice within a reasonable time frame. At the least, the manuscript should discuss in detail how the above-mentioned models differ from the presented model here.

---

## [Author Response]

Essential revisions:1. Evaluation of the (explanatory power of) the model.1a. Parameter recoverability: Please include an analysis of parameter recoverability: How well can the fitting procedure recover model parameters from data generated by the model?

We have now included a parameter recovery analysis of the MCD model. It is now included as part of the new section 3 of the revised Appendix. Importantly, our parameter recovery was performed under simulated data with similar SNR as our empirical data. In brief, the reliability of MCD parameter recovery does not suffer from any strong non-identifiability issue. However, its reliability is much weaker than in the ideal case, where data is not polluted with simulation noise.

1b. Fitting procedure: Rather than fitting the model to all dependent variables at once, it would be more compelling to fit the model to a subset of established decision-related variables (e.g. accuracy, choice confidence, choice induced preference changes) and then evaluate if, and how well, the fitted model can predict out-of-sample variables related to effort allocation (e.g. response time and subjective effort ratings). The latter would be a more stringent test of the model, and may serve to highlight its value for linking variables related to value-based decision making to variables related to meta-control.

This is an excellent suggestion. In fact, we have decided to generalize it, in the aim of providing a strong test of the model’s ability to explain all dependent variables at once. We thus performed three distinct model fits: (i) with all dependent variables, (ii) with effort-related variables only (leaving “decision-related” variables out), and (iii) with decision-related variables only (leaving effort-related variables out). We did this for each subject, each time estimating a single (within-subject) set of model parameters. We then quantified, for each dependent variable, the model’s prediction accuracy. This allows to distinguish between the accuracy of “postdictions” (i.e., the trial-by-trial correlation between data and predictions on variables that were used for fitting the model), and the accuracy of proper out-of-sample predictions (i.e., the trial-by-trial correlation between data and predictions on variables that were not used for fitting the model). Note: the latter are formally derived from parameter estimates obtained when leaving the corresponding data out. The accuracy of postdictions and out-of-sample predictions is summarized on Figure 4 of the revised Results section. In our opinion, this analysis also addresses the point 1.d below, which relates to single-subject fit accuracy (see our response below). Note that we also report group-level summaries of out-of-sample predictions for each dependent variable, when plotted against pre-choice value ratings and value certainty ratings (along with experimental data and model postdictions, see Figures 5 to 11 of the revised Results section).

1c. Model complexity: Assess (through model comparison) how many degrees of freedom are needed to account for the data (e.g. by fixing some of the crucial parameters and evaluating the fit). Currently, the authors show that their model explains more variance in dependent variables when fit to real data than random data. Almost any model which systematically relates independent variables to dependent variables would explain more variance when fit to real data than to data. It would be more useful to know whether (and if so, how much) the model explains data better, than, e.g. a model with where effort only affects precision (β efficacy), or a model in which effort only impacts value mode (γ efficacy).

We agree with you that we did not highlight explicit evidence for the existence of β and/or γ effects in the previous version of our manuscript. We have now revised our Results section to provide evidence for this. In fact, β and γ effects can be tested directly against empirical data. More precisely, under the MCD model, non-zero type #1 efficacy trivially implies that the precision of post-choice value representations should be higher than the precision of pre-choice value representations. Similarly, under the MCD model, non-zero type #2 efficacy implies the existence of spreading of alternatives. In our modified manuscript, we highlight and assess these predictions using simple significance testing on our data (see Figures 10 and 11 in the revised Results section). We note that we find this procedure more robust than model comparison in this case, given the limited reliability of parameter recovery.

1d. Single-subject data.The model appears to do fairly well in predicting aggregate, group-level data, but does it predict subject-level data? Or, does it sometimes make unrealistic predictions when fitting to individual subjects? The Authors should provide evidence of whether it can or cannot describe subject level choices, confidence ratings, subjective effort, etc.

We entirely agree with you. In the previous version of our manuscript, we had reported the accuracy of within-subject “postdictions” in the Appendix (former Figure S3). In the revised manuscript, we now report the accuracy of within-subject postdictions and out-of-sample predictions. In particular, we test for the significance of out-of-sample predictions, which provide direct evidence for the model’s ability to guess within-subject trial-by-trial variations in each dependent variable. These results are reported in the section 4.1 of the revised Results section (see Figure 4).

2. Qualify central assumptions underlying the model.2a. The model assumes that it is "rewarding" to choose the correct (highest-value) option (B = R*P). Is this realistic? If the two options have approx the same value, then R should be small (it doesn't matter which one you choose); if the options have different value, it is important to choose the correct one. Of course, the probability P_c_ continuously differentiates between the two options, but that is not the same as the reward. Can the predictions generalise toward a more general R that depends on value difference?

If you mean that people do not care about the decision when the pre-choice values are similar, then we disagree with you. In brief, we have shown that both response time and subjective effort ratings decrease when the difference in pre-choice value increases (NB: this result has been reproduced many times for RT). In other words, effort is maximal when pre-choice values are similar. This is direct evidence against the idea that decision importance (i.e., ***R*** in the MCD model) should tend to zero for such “iso-value” decisions.

Of course, decision importance is a critical component of the MCD model. This is why we had included an empirical way of manipulating it, by contrasting trials where subjects had to consume the item they chose (so-called “consequential” decisions) with trials where it was not the case (“neutral” decisions). The MCD model then predicts that people should allocate more resources (spend more time and report higher subjective effort) for “consequential” than for “neutral” decisions. In our revised manuscript, we highlight this qualitative prediction and its corresponding empirical test (cf. Figure 7 in section 4.2 of the revised Results section).

Having said this, we acknowledge that decision importance falls short of a complete and concise computational definition. In the previous version of our manuscript, we had discussed possible cognitive determinants of decision importance that would be independent of option values. Now, whether and how decision importance depends upon the prior assessment of choice options is virtually unknown. We have now modified the paragraph of the related Discussion as follows (new lines 783-815):

“First, we did not specify what determines decision “importance”, which effectively acts as a weight for confidence against effort costs (cf. in Equation 2 of the Model section). […] Probing these computational assumptions will be the focus of forthcoming publications.”

2b. Is it reasonable to assume that variance would increase as a linear function of resource allocation? It seems to me that variance might increase initially, but then each increment of resources would add diminishing variance to the mode since, e.g., new mnesic evidence should tend to follow old evidence. How sensitive are model predictions to this assumption? What about if each increment of resources added to variance in an exponentially decreasing fashion? What about anchoring biases? Because anchoring biases suggest that we estimate things with reference to other value cues, should we always expect that additional resources increase the expected value difference, or might additional effort actually yield smaller value differences over time? If we relax this assumption, how does this impact model predictions?

This is an intriguing suggestion. We recognize that, under some simple Bayesian algorithm for value estimation, one would expect some form of saturating type #2 efficacy. In other terms, the magnitude of the perturbation (per unit of resources) that one might expect when no resources have yet been allocated may be much higher than when most resources have already been allocated. We thus implemented and tested such a model. We report the results of this analysis in the section 10 of our revised Appendix. In brief, a saturating type #2 efficacy brings no additional explanatory power for the model’s dependent variables.

3. Address relationship to other accounts.3a. Does the current model predict the diverse dependent variables better than a standard accumulator models of decision-making?

This is a fair point, to which we wholeheartedly concur. We have thus implemented two simple variants of a drift-diffusion model (DDM) which can, in principle, exploit the same information as the MCD model (namely: pre-choice value difference, pre-choice value certainty, and encodings of “consequential”/”penalized”/”neutral” task conditions). We have then compared these models with MCD, w.r.t. their ability to predict out-of-sample data. The results of this comparison are reported in section 9 of our revised Appendix. In brief, standard DDM variants make quantitative predictions regarding both response times and decision outcomes, but are agnostic about choice confidence, spreading of alternatives, value certainty gain, and/or subjective effort ratings. In addition, simple DDM variants are less accurate than MCD at making out-of-sample predictions on dependent variables common to both models (e.g., change of mind).

3b. The model could also situate itself better in the broader existing literature on the topic. For instance, how does the model compares to existing computational work on this matter, e.g. the models described in Izuma and Murayama (2013) or the efficient coding account of Polanía, Woodford, and Ruff (2019)? We understand that the presented model can account for some phenomena that the other models cannot account for, at least without auxiliary assumptions (e.g. subjective effort ratings), but the interested reader might want to know how well the presented model can explain established decision-related variables, such as decision confidence, choice accuracy or choice-induced preference changes compared to existing models, by having them contrasted in a formal manner. Finally, it would seem fair to relate the presented account to emerging, more mechanistically explicit accounts of meta-control in value-based decision making (e.g. Callaway, Rangel and Griffiths, 2020; Jang, Sharma, and Drugowitsch, 2020). Ideally, some of the above would be addressed in the form of formal model comparisons, but we realise that this may be difficult to achieve in practice within a reasonable time frame. At the least, the manuscript should discuss in detail how the above-mentioned models differ from the presented model here.

This is a fair point, which we have addressed by augmenting the revised Discussion section with topic-specific paragraphs.

First, the model described in Izuma and Murayama (2013) is describing a well-known statistical artifact of measured spreading of alternatives. We have now included the following paragraph in the revised Discussion section (new lines 714-738):

“As a side note, the cognitive essence of spreading of alternatives has been debated for decades. […] Second, we have already shown that the effect of pre-choice value difference on spreading of alternatives is higher here than in a control condition where the choice is made after both rating sessions (Lee and Daunizeau, 2020).”

Second, the model by Polania and Ruff (2019) describes how limited neural coding resources shapes the transmission of information about subjective value. We comment on the relationship between this model and the MCD framework in the following paragraph of the revised Discussion section (new lines 739-759):

“A central tenet of the MCD model is that involving cognitive resources in value-related information processing is costly, which calls for an efficient resource allocation mechanism. […] A possibility is to consider, for example, energy-efficient population codes (Hiratani and Latham, 2020; Yu et al., 2016), which would tune the amount of neural resources involved in representing value to optimally trade information loss against energetic costs.”

Third, the models of Callaway et al. (2020) and Jang et al. (2020) effectively consider optimal policies for dividing attention between items in the choice set. They are very similar to each other, although the work by Jang et al. (2020) has a more solid theoretical grounding. We thank you for pointing us to these papers, which we were not aware of. We now refer to these work in the following modified paragraph of the Discussion section (new lines 824-830):

“More problematic, perhaps, is the fact that we did not consider distinct types of effort, which could, in principle, be associated with different costs and/or efficacies. […] Such optimal adjustment of divided attention might eventually explain systematic decision biases and shortened response times for “default” choices (Lopez-Persem et al., 2016).”